# Production of individualized V gene databases reveals high levels of immunoglobulin genetic diversity

Martin M. Corcoran[1],*, Ganesh E. Phad[1],*, Néstor Vázquez Bernat[1], Christiane Stahl-Hennig[2], Noriyuki Sumida[1], Mats A.A. Persson[3], Marcel Martin[4] & Gunilla B. Karlsson Hedestam[1]

Comprehensive knowledge of immunoglobulin genetics is required to advance our understanding of B cell biology. Validated immunoglobulin variable (V) gene databases are close to completion only for human and mouse. We present a novel computational approach, IgDiscover, that identifies germline V genes from expressed repertoires to a specificity of 100%. IgDiscover uses a cluster identification process to produce candidate sequences that, once filtered, results in individualized germline V gene databases. IgDiscover was tested in multiple species, validated by genomic cloning and cross library comparisons and produces comprehensive gene databases even where limited genomic sequence is available. IgDiscover analysis of the allelic content of the Indian and Chinese-origin rhesus macaques reveals high levels of immunoglobulin gene diversity in this species. Further, we describe a novel human IGHV3-21 allele and confirm significant gene differences between Balb/c and C57BL6 mouse strains, demonstrating the power of IgDiscover as a germline V gene discovery tool.

[1] Department of Microbiology, Tumor and Cell Biology, Karolinska Institutet, Stockholm SE-17177, Sweden. [2] Deutsches Primatenzentrum GmbH (DPZ), Leibniz-Institute for Primate Research, Göttingen D-37077, Germany. [3] Department of Clinical Neuroscience, Stockholm University, Box 1031, Solna SE-17121, Sweden. [4] Science for Life Laboratory, Department of Biochemistry and Biophysics, Stockholm University, Box 1031, Solna SE-17121, Sweden. * These authors contributed equally to this work. Correspondence and requests for materials should be addressed to M.M.C. (email: Martin.Corcoran@ki.se) or to M.M. (email: Marcel.Martin@scilifelab.se) or to G.B.K.H. (email: Gunilla.Karlsson.Hedestam@ki.se).

The adaptive immune response is dependent on the selection of mature B cells expressing antigen-specific antibodies from a diverse repertoire of naive B cells[1,2]. In recent years, the advent of next-generation sequencing (NGS) technologies have provided new opportunities to examine expressed antibody repertoires in both human and model species, forging new insights into how B cells respond to, and are shaped by, external stimuli[3]. These analyses involve the comparison of expressed antibody sequences with reference databases of variable (V) germline segments to determine gene usage, expression frequency and degree of somatic hypermutation (SHM), among other genetic features. This requirement for accurate and complete immunoglobulin (Ig) gene reference databases[4], however, severely curtails the widespread use of antibody repertoire analysis. Although partial V gene databases exist for many species, relatively complete germline Ig reference databases are currently available only for human and mouse[5] and even these may not be as comprehensive or correct as previously assumed. Importantly, knowledge of germline sequences in a given species is particularly necessary for applied approaches, for example, providing the ability to design amplification primers for high-throughput cloning of paired heavy and light chains to isolate antibodies of potential therapeutic value.

Recent studies demonstrate that computational and screening approaches can identify novel, rare human and mouse V alleles[6,7]. However, a reliable procedure to construct a germline V gene database de novo remains elusive, in particular for species that lack relatively complete reference genomes. Here we describe a novel computational approach to define germline V sequences within NGS data to a level that enables individualized database construction. IgM antibody libraries contain a mixture of naive germline V sequences in addition to those subjected to SHM, with both groups exhibiting additional low-rate sequence variation introduced by PCR or sequencing errors. We demonstrate here that germline V gene sequences can be defined from this mixture by identifying clusters within groups of sequences assigned to a rough 'initial' database. Consensus sequences, produced from these clusters, represent candidate germline sequences as shown using a computational screening procedure that retains germline sequences but removes false positives. We have automated these steps in one single application named IgDiscover. We validate this approach by (i) successfully re-discovering human VH alleles starting from an artificially reduced database, (ii) identifying the same sequences expressed in several individual animals and (iii) by direct cloning of newly identified sequences from non-rearranged genomic DNA. We further demonstrate that the approach can produce complete germline V gene databases for each individual tested. Finally, we show that germline V gene repertoires differ considerably between individual animals used for immunization studies, highlighting both the need to create accurate databases specific to each individual studied and demonstrating the utility of IgDiscover as a means to achieve this goal.

## Results

**V gene database assembly.** The availability of a complete data-base of V gene segments for a given species is the exception rather than the norm. Ig loci are repetitive and difficult to assemble. In only a few cases, such as humans and commonly used mouse strains, the loci are sequenced without gaps and the number of V genes is known[8,9]. Without a high-quality reference genome, gaps in the sequence typically result in an incomplete list of known V segments (Fig. 1a).

In addition, rare alleles exist in some individuals that are not present in the reference database. The total number of V alleles present within any given species is dependent on the genetic diversity of the population[10]. Currently, the number of sequences denoted as functional VH alleles present within the IMGT database, the most comprehensive resource of curated Ig sequences[11], are 254 and 238 for human and mouse, respectively (August 2016) (Fig. 1b). Even within these species, however, comprehensive analyses of diverse populations are lacking and the total number of V heavy and light germline alleles is likely to be higher[12]. To facilitate the identification of germline alleles, we sequenced libraries of IgM heavy chain V, diversity and junction (J) transcripts in multiple species and developed a versatile V gene discovery software, IgDiscover. Germline sequences identified by IgDiscover can be either genes or allelic variants. The program is designed to identify differences between expressed germline sequences rather than identifying whether an individual sequence is a gene or allele and we use these terms interchangeably.

**Discovery method overview.** The method consists of four critical steps: initial assignment to a starting database; clustering to identify candidate germline sequences; and application of a filter to retain only true germline sequences that are output as a new database that replaces the starting database. Finally, the previous three steps are repeated using the replaced database as the new assignment database. IgM libraries are utilized in order to minimize SHM levels and 5′ rapid amplification of cDNA ends (5′RACE) amplification used to amplify sequences from species with limited genomic information available. For other species, either 5′RACE or multiplex PCR can be used for library amplification. IgDiscover requires a starting database of either VH or V-kappa (VK) or V-Lambda (VL) genes that are used for primary assignment. The program updates the database incrementally over the course of multiple iterations to produce a final database of expressed germline V genes that are specific to the individual (Supplementary Fig. 1). These steps are described in detail below.

**Identification of novel VH allele candidates.** At the beginning of each iteration, the V, diversity and J genes in the input sequences are localized and classified using IgBLAST[13]. A key observation is that any novel germline allele is assigned to the most similar database sequence. IgDiscover therefore inspects database alleles one by one in order to find clusters of novel sequences. When no novel sequences are present, that is, the database germline allele is identical to the expressed allele, the percentage differences appear Poisson-distributed (left histograms in Fig. 2a). Novel germline genes appear in three different ways. First, for a single novel allele, if the closest database sequence is not expressed in that individual, the distribution of percentage differences is shifted to the right, and there are few or no exact matches (Fig. 2a, right histogram). To cover this case, IgDiscover computes the candidate novel allele as a consensus of all sequences represented in the plot. The second case is that both the database allele and the novel germline allele are expressed and assigned to the same database gene. This results in a combined, bimodal distribution such as in Fig. 2b (right histogram), where the left cluster corresponds to the known database germline allele and the right cluster to the candidate novel allele.

In the third case, several novel germline alleles are assigned to the same database allele, but their peaks overlap in the histogram and can therefore not be resolved (Fig. 2c,d). IgDiscover handles this case by also clustering sequences according to their similarity to each other. The sequences assigned to the database allele are clustered hierarchically using the UPGMA algorithm[14]. In the graphical representation of the result, Fig. 2f, subclusters of

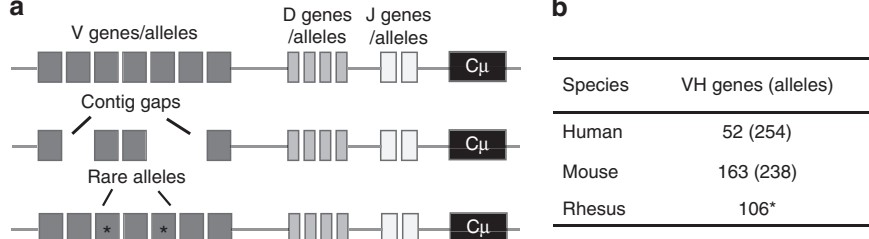

**Figure 1 | IgH genomic locus. (a)** Issues affecting VH database construction based on genomic reference assemblies. The top locus map contains a fully sequenced and assembled IgH region such as found in the human and mouse reference assemblies. The centre map illustrates genome assemblies with insufficient coverage resulting in the absence of some VH genes in the reference genome. The lower map illustrates the presence of rare genes/alleles that are yet to be identified in genetically diverse species. **(b)** Number of VH genes and alleles in human and mouse that are denoted as functional according to the IMGT database and the available set of published rhesus VH genes (August 2016) compiled from published sources[31,43,44]. *Rhesus sequences may be either genes or alleles.

similar sequences appear as light squares along the diagonal. They are detected by IgDiscover through a heuristic (see Methods), and consensus sequences computed from each subcluster are used as additional candidate novel VH alleles. We refer to this approach as linkage cluster analysis.

**Identifying 'missing' germline V genes from a human database.** As a novel germline sequence will be recognized as different to the set of reference germline sequences in a given screening database, we surmised that it would be possible to model the expression of a novel allele by removing the sequence of known germline alleles from the screening database. This is illustrated by the re-identification, by IgDiscover, of single (Fig. 2a,b) or multiple (Fig. 2e,f) database-deleted alleles in a human IgM library we constructed using 5′RACE. Although a minority of alleles could be retrieved using the windowed clustering approach, the discovery of the most sequences required combining windowed and linkage cluster analysis. A single iteration of IgDiscover resulted in the re-identification of six of the seven expressed alleles that had been removed from the reference database (Fig. 2f).

**Germline filter.** Initial candidate germline V sequences may include false positives, in particular mutation hotspot artefacts and mutations arising from PCR bias. We employ a germline filter (Fig. 3) after the last iteration to remove them.

A pregermline filter is used in earlier iterations. It is less strict, which aids in splitting up highly complex initial assignments, for example, in the case of VH families that contain many family members, that is, the VH3 or VH4 families in primates or the VH1 family in mice. It also accelerates the identification of germline alleles from small clusters. The germline filters test features present only in germline genes, in particular whether the candidate sequence occurs in association with multiple HCDR3 sequences, thus demonstrating independent rearrangements (see Methods). The use of the germline filter therefore results in a final database of high specificity. In order to test this, we analysed a human IgM library with IgDiscover. In this case, the proportion of validated germline sequences reached 97% after germline filtering (Fig. 3c).

**IgDiscover processes data in an iterative manner.** Following library production, sequencing, preprocessing and initial assignment to a starting database of V sequences (Fig. 4a,b), IgDiscover identifies clusters using the windowed and linkage cluster techniques (Fig. 4c)

Consensus sequences that contain no ambiguous bases are screened using the pregermline filter and the set of candidate

germline sequences that pass are used as a new V germline database. The process is repeated using this new V database as the next assignment database.

The initial database is replaced by a new database of candidate germline V alleles identified from the cluster analysis and filtering process. These new database sequences, as they are based on common sequences expressed within the library, function to create a new set of assignments in the next iteration, that, in turn, lead to an accelerated identification of additional germline candidates. The precise sequence of the starting database is not critical; a database where each sequence has 10 random mutations introduced results in a similar output to a non-mutated database (Fig. 5b). Initial assignment to the database sequences by IgBLAST requires some degree of sequence similarity. The starting database therefore must consist of V gene sequences; however, the species from which the database is constructed can be different to the library being analysed. Both human and mouse databases can function efficiently as starting databases for analysis of the rhesus library (Fig. 5c,d).

Because linkage cluster analysis is randomized (see Methods), it is important that repeated analyses of a single IgM library identify the same set of germline sequences. Analysis of Chinese-origin rhesus F132 on separate occasions consistently produced 70 germline sequences. IgDiscover is designed to identify a database of expressed germline sequences independent of the composition of the starting reference database content. This is reflected in the results of a series of tests comparing the final outputs of IgDiscover analysis of Chinese-origin rhesus F132. A final VH database of the same 60–70 alleles was produced when IgDiscover was run using a wide variety of starting databases using 3 iterations. These included: a validated set of 106 Indian-origin rhesus VH sequences; the same set of sequences that have each been randomly mutated by 10 nucleotides; starting databases consisting of only human VH sequences or only mouse VH sequences; and finally a starting database of just a single rhesus VH gene, exemplified here by using VH7.21 (Fig. 5a–e).

**Validation of identified germline sequences.** We applied IgDiscover to an IgM library from an Indian rhesus macaque no. 2635, using a 'minimal' starting database, consisting of a single gene from each of the seven primate VH families. IgDiscover identified 53 germline alleles (Fig. 6a).

Two initial validation techniques were used to ensure that the sequences identified by IgDiscover were true germline alleles. The first method involved the identification of rhesus macaque genomic sequences or previously validated germline alleles, that shared 100% identity to the 53 germline sequences. In all, 35/53 (66%) were 100% identical to sequences present in the Indian

rhesus reference genome (Mmul_8.01 reference assembly) and/or other previously published rhesus V alleles.

The second technique utilized the approach described by previous groups and involved identifying the presence of the same sequence in more than one individual[7,15]. Of the 35 sequences that shared identity with germline sequences, 28 were present in at least one of the three additional monkeys analysed (rhesus nos 2514, 2636 and 5200), indicating that this approach could confirm germline alleles. Of the 18 novel alleles, 13 were present in one or more of the additional animals (Fig. 6a). Examination of the frequency of the 35 previously published germline VH alleles and the 18 novel alleles in the expressed IgM

repertoire of this animal revealed that many of the novel alleles were utilized to a similarly high frequency as the known alleles (Fig. 6b).

**Genomic cloning of unrearranged germline VH sequences.** In contrast to the relatively well-validated genomic information available for Indian-origin rhesus VH sequences, no validated VH database from Chinese-origin rhesus macaques has been published to date. We therefore sought to perform genomic validation on a representative set of sequences identified in Chinese-origin rhesus macaques by IgDiscover. We designed amplification primers encompassing previously validated Indian-origin rhesus V gene segments. Indian- and Chinese-origin rhesus macaques are highly similar at the genomic sequence level[16]. Genomic primers encompassing VH segments, based on Indian-origin rhesus genomic sequence, may therefore be expected to amplify the same VH segments from Chinese-origin rhesus genomic DNA. Over the course of evolutionary time, VH gene families both expand and contract owing to genomic duplication and deletional events[10]. Genomic PCR primers targeting known VH alleles may therefore additionally co-amplify related VH sequences that have arisen owing to gene duplication or the presence of copy number variants[9].

Following PCR amplification, we cloned 42 germline sequences that were identical to the corresponding sequences identified by the IgDiscover program from Chinese-origin rhesus no. F124 genomic DNA. Of these, 34 were novel alleles, including 3 novel VH2 family alleles (Fig. 7a,b), and 8 were identical to previously published VH germline sequences from the Indian-origin database. Some primer sets that encompassed known VH genes amplified more than two germline sequences from rhesus no. F124. Primers (Supplementary Table 1) encompassing VH.34,

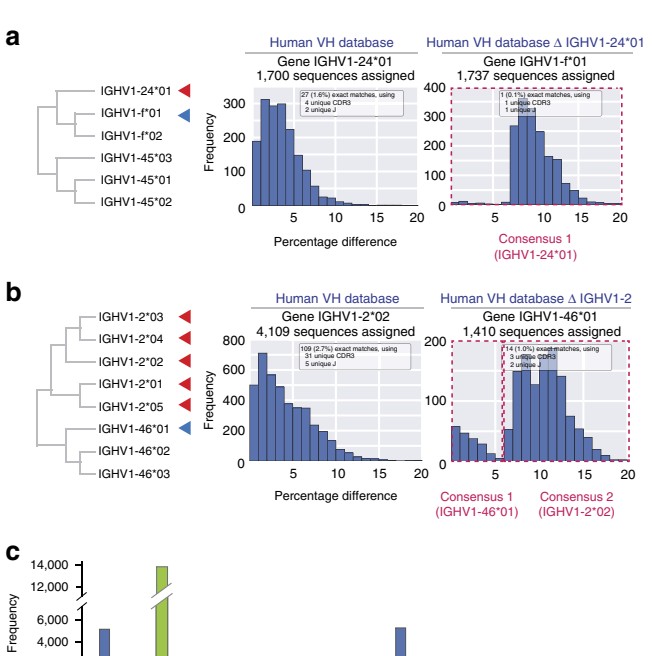

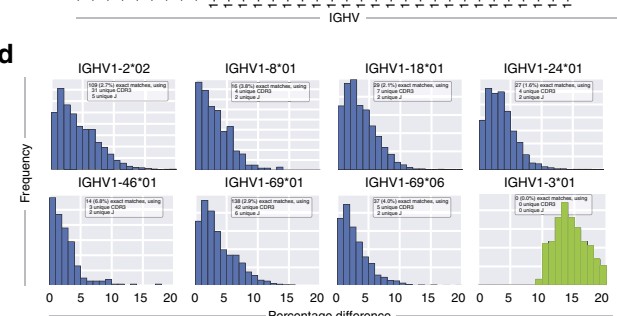

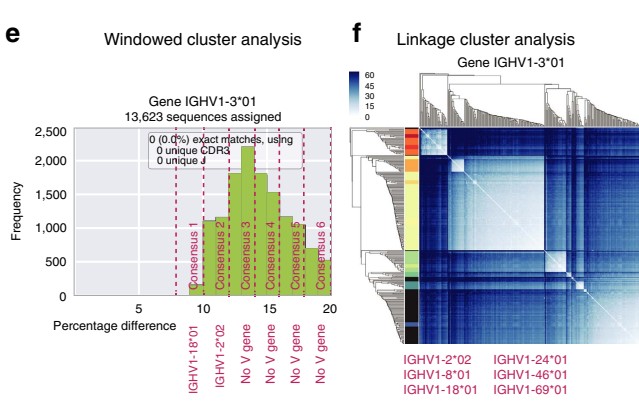

**Figure 2 | Identification of human VH alleles when the human reference database is rendered incomplete.** Phylogenetic tree of relationship between (**a**) IGHV1-24*01 and IGHV1-f*01 or (**b**) the IGHV1-2 alleles and IGHV1-46*01, where the red arrow shows the deleted allele and the blue arrow shows the closest allele to which the sequences are now assigned. Frequency plot of sequences assigned to (**a**) IGHV1-24*01 (centre panel) or (**b**) IGHV1-2*02 (centre panel) when the reference database is complete. The right panels shows the new assignment when IGHV1-24*01 (**a**) or IGHV1-2 alleles (**b**) are deleted from the reference database. The pink text below the right panels indicate the identification of one consensus sequence from the 1–20% range, identical to IGHV1-24*01 (**a**) or two consensus sequences from 1–6% and 6–20%, showing 100% identity to IGHV1-46*01 and IGHV1-2*02, respectively (**b**). (**c**) Bar chart showing the assignment of IGHV1 human IgM sequences using the complete starting database (blue bars) or an incomplete database from which all IGHV1 alleles apart from IGHV1-3*01 have been removed (green bars). (**d**) Frequency plots of the seven IGHV1 alleles expressed in the human IgM library (blue histograms) and the resultant frequency plot after all IGHV1 alleles apart from IGHV1-3*01 have been removed (green histogram). (**e**) Frequency plot of sequences assigned to IGHV1-3*01 showing Windowed cluster analysis. The assigned sequences are binned in 2% intervals and from each a consensus sequence is built. The pink text beneath the panel reveals the identification of consensus sequences, from the 8–10% and the 10–12% windows, showing 100% identity to IGHV1-18*01 and IGHV1-2*02, respectively. (**f**) Matrix diagram showing Linkage cluster analysis of sequences assigned to IGHV1-3*01 following deletion of the other IGHV1 alleles from the VH screening database. Clusters of closely related sequences are visualized as lighter coloured squares within a darker blue background. The coloured bars on the left of the figure show the numbers of clusters identified in this analysis. The red text beneath the panel reveals that consensus sequences from the linkage clusters were identical to six of the seven IGHV1 sequences.

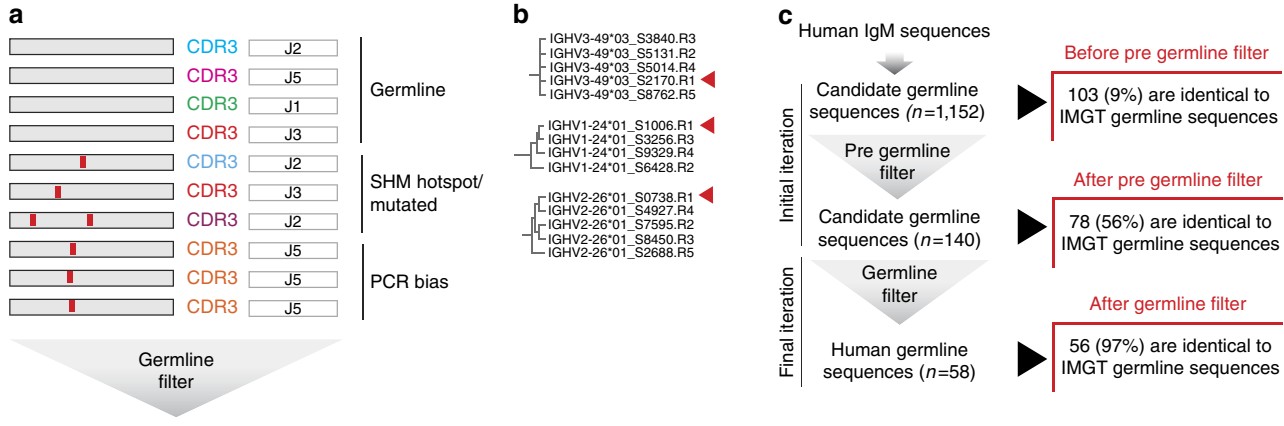

**Figure 3 | Germline filter.** (**a**) Schematic of types of sequences present following clustering and consensus building. True germline sequences are found in antibody sequences that share an identical V region and are present in multiple unique rearrangements as shown by a high number of unique CDR3 segments. Non-germline sequences include those that have undergone SHM at activation-induced cytosine deaminase (AID) hotspots and, finally, those that are overrepresented owing to PCR bias issues. The application of the germline filter leaves only germline sequences remaining in the final output. (**b**) Illustration of clustal analysis of a section of candidate germline VH alleles from the H1 human IgM library before germline filtering. Each sequence was compared with the full IMGT human VH database and those identical to germline genes are indicated with a red arrow. The relative frequency of V usage, based on unique CDR3 content, is indicated by the number following the 'R' of each sequence cluster (R1 being the highest frequency). (**c**) Results of the application of the pregermline and germline filters following a single iteration analysis of the H1 human IgM library. Candidate VH sequences ($n = 1152$), were produced via consensus building using clusters identified by Windowed and Linkage cluster analysis (see Methods). To illustrate the effect of the germline filter sequences were screened against the current (August 2016) IMGT human VH database to identify the proportion showing 100% identity to validated IMGT germline sequences. This analysis was repeated with the resulting outputs of the pregermline and finally the germline filter.

VH4.28, VH4.26 and VH3.15 amplified three germline sequences while VH4.11 amplified four (Table 1), indicating prior VH gene duplication and expansion.

**IgDiscover production of individualized databases.** *Human.* Individualized germline databases of three human IgM libraries, H1, H2 and H3, were produced using IgDiscover. The numbers of alleles detected for each library were 58, 66 and 60, respectively (Supplementary Data). All known functional VH genes were detected by IgDiscover with the exception of just two genes: IGHV3-NL1*01, a sequence isolated from an individual from the Papua New Guinea Highlands, and the sequence IGHV3-72*01. This latter sequence is possibly non-functional as there is a lack of antibody sequences with unmutated full-length IGHV3-72*01V segment in the NCBI nr database, and previous studies of multiple individuals has failed to detect expression[17].

Of the three human repertoires, most sequences (56/58 for H1, 64/66 for H2 and 56/60 for H3) were identical to known germline sequences from the current IMGT database (Supplementary Fig. 2). Two novel sequences were found in all three personalized repertoires, one of which, termed IGHV3-21_DEL, a novel variant of the gene IGHV3-21, was validated by genomic cloning (Fig. 7c).

*Mouse.* Three mouse IgM libraries were sequenced, two M1 and M2 from Balb/c strain mice and one M3, from C57BL6. In addition, an archived C57BL6 IgM library, produced by Ion Torrent sequencing and deposited to the public NCBI sequence read archive (SRA SRX1794884), was retrieved and analysed by IgDiscover. A total of 214 unique germline sequences were identified (Supplementary Data, Supplementary Fig. 3). The results indicate that the Balb/c mice were strikingly similar, sharing a total of 87 identical alleles between the two mice (M1 = 122 sequences, M2 = 105 sequences). In contrast, the two C57BL6 mice libraries contained 44 identical sequences (database size of 71 and 57 sequences for the M3 library and the Ion

Torrent C57BL6 library, respectively) but shared just 10 identical sequences with the Balb/c mice.

*Rhesus.* IgDiscover analysis was performed on IgM libraries from five rhesus macaques of Chinese origin. The analysis utilized either a seven sequence 'minimal' starting database that contains a single allelic sequence for each of the seven VH families (Fig. 8a) or the public rhesus macaque VH database (Fig. 8b). IgDiscover analysis of five Chinese-origin macaques produced a combined database of 240 unique germline VH sequences (Fig. 8b), of which 30 were identical to VH alleles present in the Indian-origin rhesus database. The average number of VH alleles detected per monkey was 67. Although many of the alleles were present in more than one animal, all individuals analysed contained multiple alleles that were not shared by any of the other four animals (Fig. 8c), demonstrating the high genetic diversity of this locus and the importance to properly determine the V gene germline composition in individual subjects before subsequent antibody sequence analysis.

**Light chain V gene discovery.** The principles of germline V gene discovery that are used to identify VH germline sequences, namely, initial assignment, cluster identification and application of the germline filter, were also applied to the discovery of germline VK and VL light chain sequences in three Chinese-origin rhesus macaques. The results (Fig. 8d,e) reveal a high degree of diversity, with only a minority of VK alleles (30/127), and VL alleles (18/135), present in more than one of the three individual monkeys. As light chain sequences obtained by RACE PCR will consist of a mixture of both naive sequences and those that have undergone SHM in the germinal centres, we tested the ability of IgDiscover to identify germline sequences from a library consisting of a mixture of an IgM and an IgG library from a single rhesus individual. The results, in Fig. 5g, show that when IgDiscover is applied to the mixed library, it identifies the majority of germline sequences (68/69) that are present in the

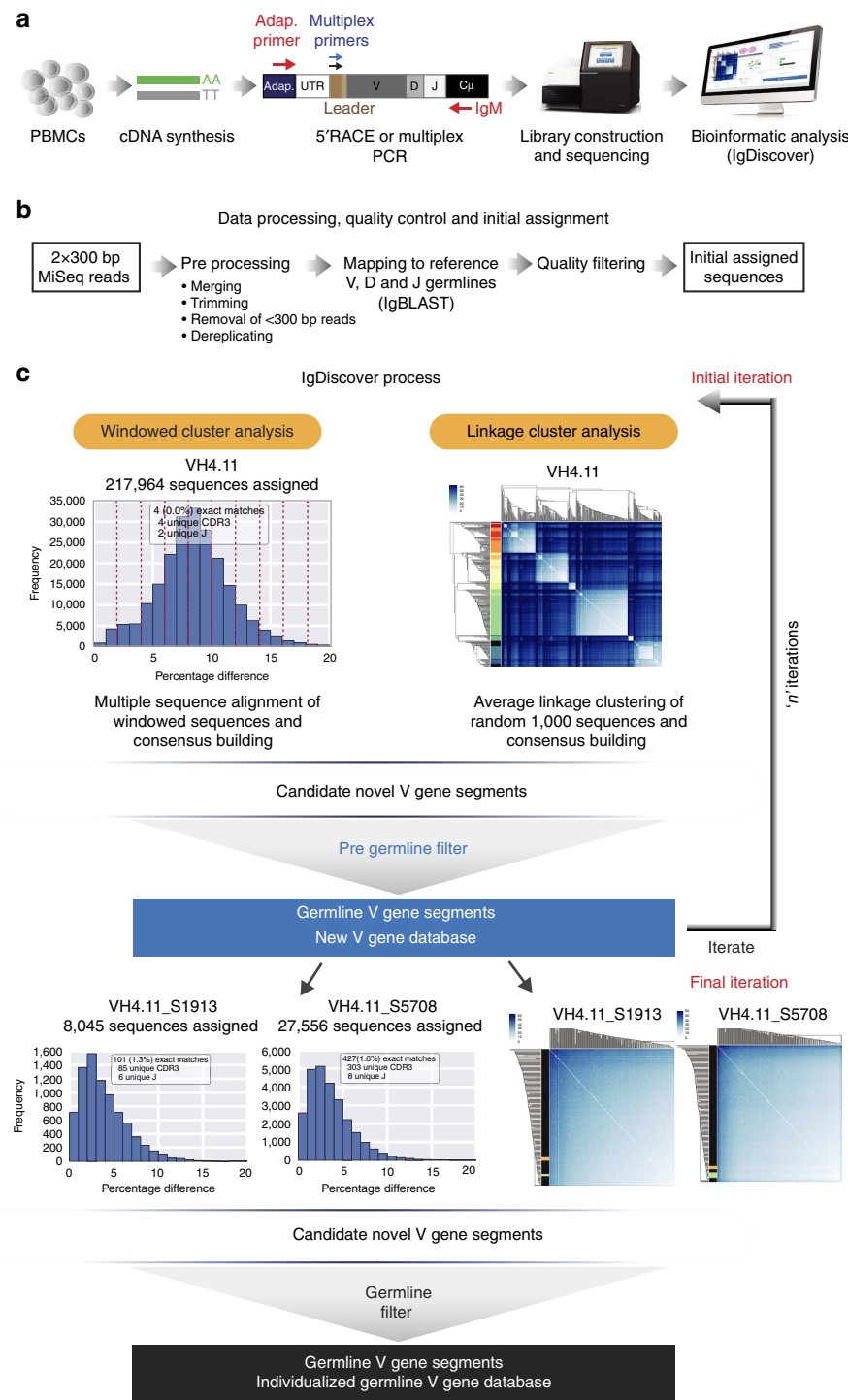

**Figure 4 | The IgDiscover iterative gene discovery process.** (**a**) Following isolation of lymphocyte messenger RNA, an IgM library is constructed using either 5′RACE or multiplex PCR and sequenced using the Illumina MiSeq system. (**b**) Paired sequences are merged, and adapters are optionally removed. IgBLAST then assigns VH segments based on the starting database, and low-quality assignments are filtered (see Methods). (**c**) Windowed cluster analysis, (upper left panel), showing sequences assigned to a reference gene and binned in 2% windows to allow discrete consensus building. Linkage cluster analysis (upper right panel) of a subset of 300 sequences. Rows and columns of the matrix correspond to sequences. The colour at an intersection of a row and a column gives the number of differences between the corresponding sequences. The dendrograms (to the left and above, both identical) show the hierarchy found according to hierarchical clustering. Sequences are rearranged to conform to the clustering, putting similar sequences adjacent to each other. Clusters of similar sequences are visible as bright squares along the main diagonal. Colouring on the left indicates clusters detected by IgDiscover (one colour per cluster). Following consensus building, candidate germline sequences are processed using the pregermline filter, resulting in a new VH database. The assignment, clustering, consensus-building and pregermline filter steps are repeated for a set number of iterations using each new VH database for initial assignment. Examples of windowed cluster histograms (left two lower panels) and linkage cluster plots (right two lower panels) using databases containing newly identified candidate germline V genes. In the final iteration, the candidate V genes are processed using the germline filter to reveal the final database.

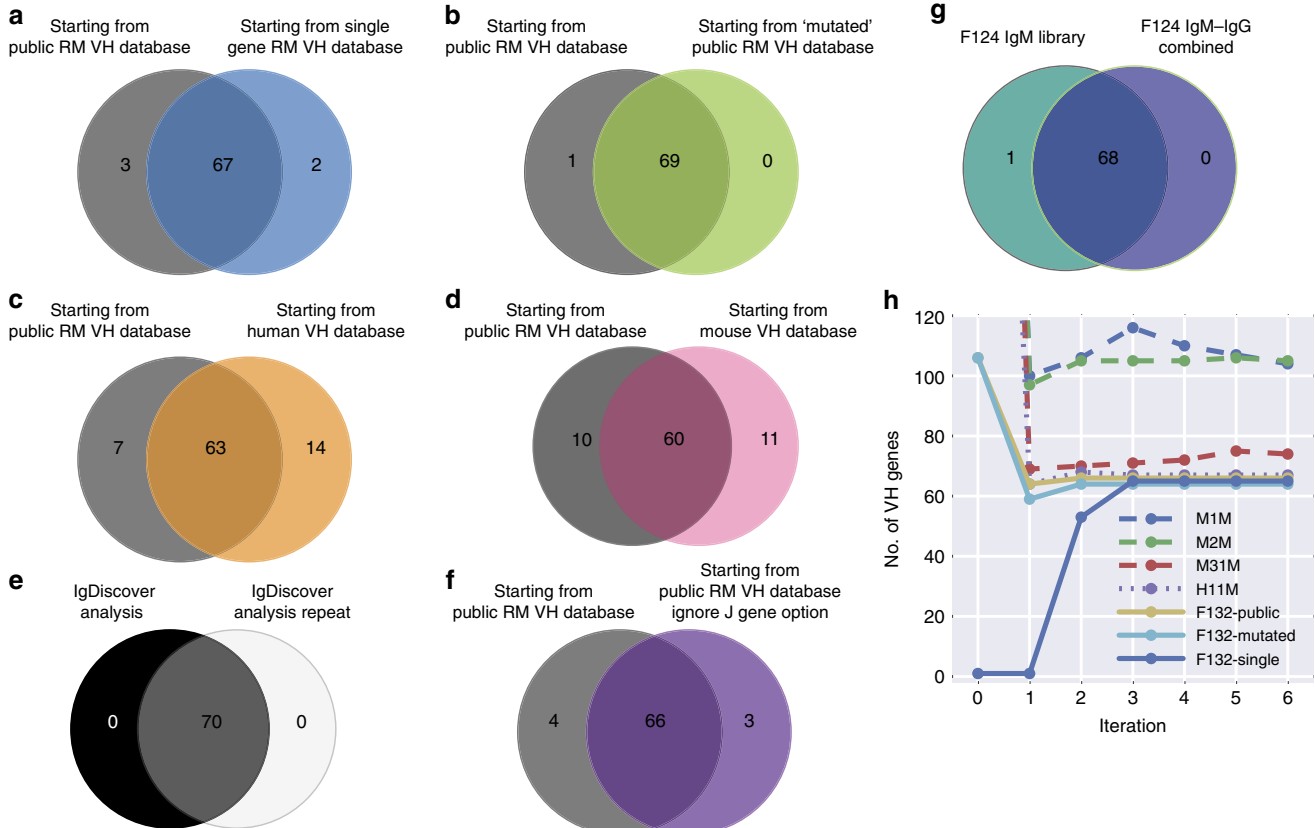

**Figure 5 | Starting database effect on final output and iteration requirement.** Venn diagrams comparing the number and identity of alleles output by IgDiscover when the following series of starting databases or conditions are utilized: (**a**) Public RM VH database constructed from previously published rhesus germline V sequences[19,43–45] versus 'Single gene RM VH' database, a VH database containing a single sequence, (VH7.21). (**b**) Public RM VH database versus 'Mutated Public RM VH database', a database created by randomly mutating 10 bases in every sequence within the Public RM VH database. (**c**) Public RM VH database versus the Human database, VH database created using 351 sequences from the IMGT collection of human VH sequences, none of which are identical to rhesus macaque VH sequences. (**d**) Public RM VH database versus the Mouse database, VH database created using 407 sequences from the IMGT collection of mouse VH sequences, none of which are identical to rhesus macaque VH sequences. (**e**) Public RM VH database versus duplicated analysis of Public RM VH database. (**f**) Public RM VH database using standard J gene settings versus the Public RM VH database when J gene identity is not part of the filtering criteria. (**g**) Comparison of germline VH alleles identified from F124 IgM library compared with germline VH alleles identified from a combined IgG–IgM library data set. (**h**) Illustration of the number of iterations to produce a final database from mouse (M1M, M2M, M3M), human (H1M) and rhesus (F132). Each dot indicates the size of the individualized database if the process had been stopped at that iteration and the germline filter applied.

IgM library, indicating that the IgDiscover clustering and filtering procedure enables identification of germline sequences even when the library to be analysed contains only a small proportion of germline sequences.

**5′ untranslated region (UTR) discovery for primer design**. As the IgDiscover protocol utilizes 5′RACE to achieve an unbiased amplification of V genes, it enables the identification of antibody messenger RNA upstream of the V gene segment. The IgDiscover program contains a script to extract this information in table form for each novel sequence identified (Table 2).

## Discussion

The ability of IgDiscover to create individualized V gene databases *de novo* facilitates a new level of analysis in B cell research. It provides, primarily, a definitive reference for each subject analysed, a critical step for the identification of specific germline alleles utilized by individual antibodies and for determining their levels of SHM. Immunological diversity at the

level of Ig alleles is now recognized as a factor of some importance[12]. Studies involving different ethnic groups have revealed novel human VH alleles[15,18] and comprehensive screening procedures may uncover many additional examples. Additionally, the availability of comprehensive Ig databases for laboratory animals commonly used in immunological research would enable antibody analysis to be extended to these species. However, while a technique to automate the identification of novel alleles has been developed[6], it currently requires a relatively complete starting database and so is limited to species in which such a resource is available. In addition, it is sensitive to the issue of gene duplication and allelic copy number diversity, a factor that our genomic validation results indicate to be relevant (Table 1). IgDiscover was developed to overcome these limitations.

Critically, IgDiscover analysis simplifies both inter- and intra-species comparison as shown here where we demonstrate a high degree of allelic diversity within Indian- and Chinese-origin macaques, two groups of non-human primates frequently used as models for human biology. The level of macaque Ig allelic diversity revealed is striking; however, it is consistent with a

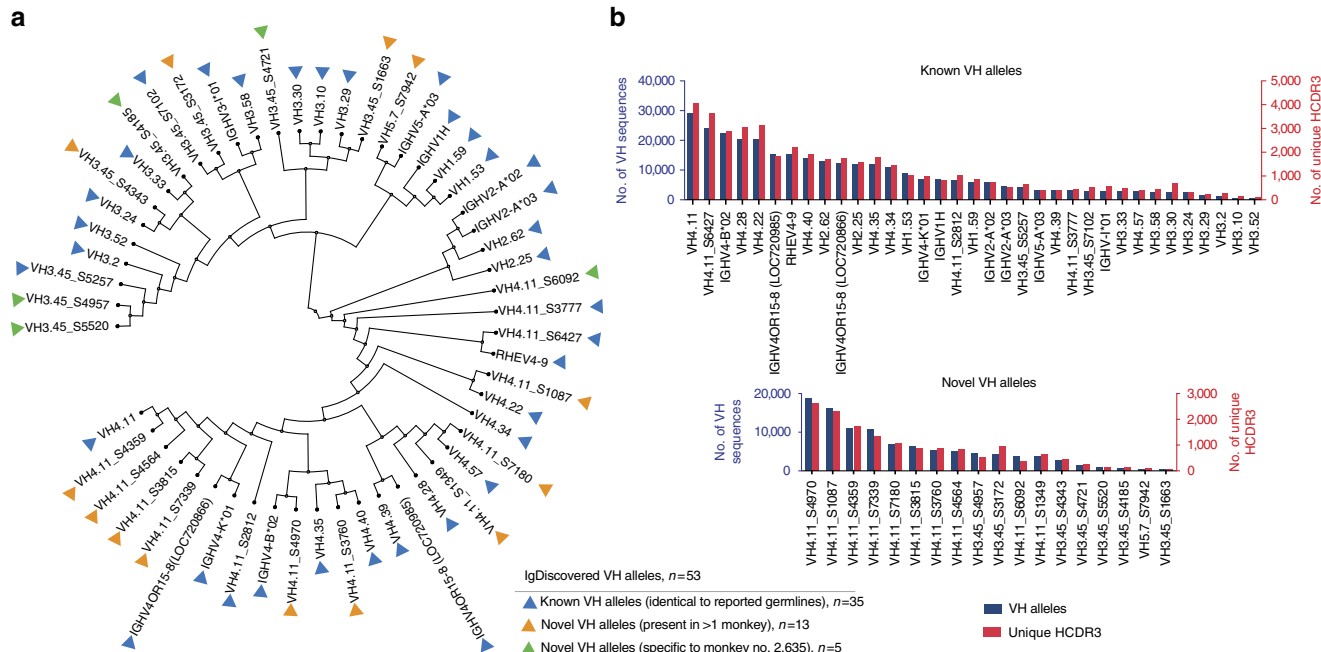

**Figure 6 | Analysis of novel VH sequences identified by IgDiscover in an Indian-origin rhesus macaque.** (**a**) Phylogenetic tree of VH alleles ($n = 53$) from rhesus macaque no. 2635, analysed using a minimal seven gene starting database (VH1.53, VH2.12, VH3.45, VH4.11, VH5.7, VH6.1 and VH7.21). The resulting germline sequences show 100% identity to rhesus genomic reference sequence or to previously published germline alleles (blue arrows, $n = 35$), 100% identity to germline alleles present in additional Indian rhesus animals (orange arrows, $n = 13$) or present only in rhesus macaque no. 2635 (green arrows, $n = 5$). (**b**) Frequency of the usage of novel alleles. Panels showing the results of V sequence assignment of rhesus macaque no. 2635 using the database created by IgDiscover. The blue columns represent the number of sequences assigned to the database allele, while the red columns represent the numbers of unique CDR3s associated with that allele. The upper chart represents alleles that have 100% identity to rhesus genomic reference sequence or previously published germline alleles. The lower chart includes only those alleles that have not been published previously and are defined as novel rhesus germline alleles.

recent study by Collins *et al.*[7] comparing laboratory mouse strains Balb/c and C57BL/6 which showed that just five alleles were common to both strains.

To ensure the accuracy of the individualized databases, considerable efforts have been applied to the validation of the novel alleles discovered during the development of this tool. Germline alleles uncovered by IgDiscover have been confirmed through identification in multiple individuals or the presence in the reference genomic assembly (Fig. 6a,b) and finally through genomic cloning of unrearranged sequences (Fig. 7a,b). The level of inter-individual diversity among both VH and VK/VL alleles identified suggests that previous estimations of VH repertoire usage and SHM levels utilizing earlier databases constructed from single animal reference genomes are likely to include errors. Tackling this issue in future studies could involve the use of more comprehensive germline databases than is currently available. Such resources could be constructed from the V gene sequences of large number of individuals in order to ensure that close to all alleles are included. This is especially important for species where the diversity is very high, as observed here for rhesus macaques. We can contribute to this endeavour with the 279 allele database of Chinese-origin VH sequences discovered in the current study (Fig. 8b,c), in addition to increasing the rhesus VK and VL databases by 102 and 21 allelic sequences, respectively. Alternatively, a preferred approach would be to construct individualized databases for each subject (animal or human) analysed. Given the increased accessibility of NGS today, this is readily achievable using IgDiscover and requires the sequencing of a single IgM library and not more than a few hours of computing time for VH analysis and a similar timescale for VK and VL analysis.

Importantly, IgDiscover requires minimal prior sequence information for efficient V gene discovery. The starting database sequences are not required to be identical to any germline present in the test case. We demonstrate here that by using starting databases consisting of Indian-origin rhesus germline alleles, deliberately mutated rhesus alleles or even databases of human or mouse alleles, IgDiscover constructs databases consisting of the exact same 75–79 alleles of rhesus germline VH sequences (Fig. 5b–d). In an extreme test case, even a single VH sequence was shown to function efficiently as a starting 'database' to identify the same set of germline alleles (Fig. 5a). The program is therefore applicable to the construction of VH databases for species where little genomic sequence is currently available. Studies of B cell biology in non-human or mouse contexts are, to date, less common, in part owing to a lack of familiarity with the genetics of other model organisms[10,19,20]. Extending antibody repertoire analysis to multiple new species, however, will expand the possibilities for immunological research of relevance for studies of human pathogens, such as those currently performed in voles (hantavirus)[21], ferrets (influenza virus)[22] or different non-human primate species (Ebolavirus and many other infections)[23]. The use of IgDiscover in diverse species may also uncover species-specific evidence of past exposure to pathogens, aiding the elucidation of disease susceptibility and resistance.

The ability to analyse antibody repertoires in additional species also introduces the possibility of isolating monoclonal antibodies of particular specificities produced in such animals. IgDiscover includes a capability to identity the 5′UTR and leader sequence upstream for each V allele identified (Table 2), therefore providing the means to design amplification primers for high-

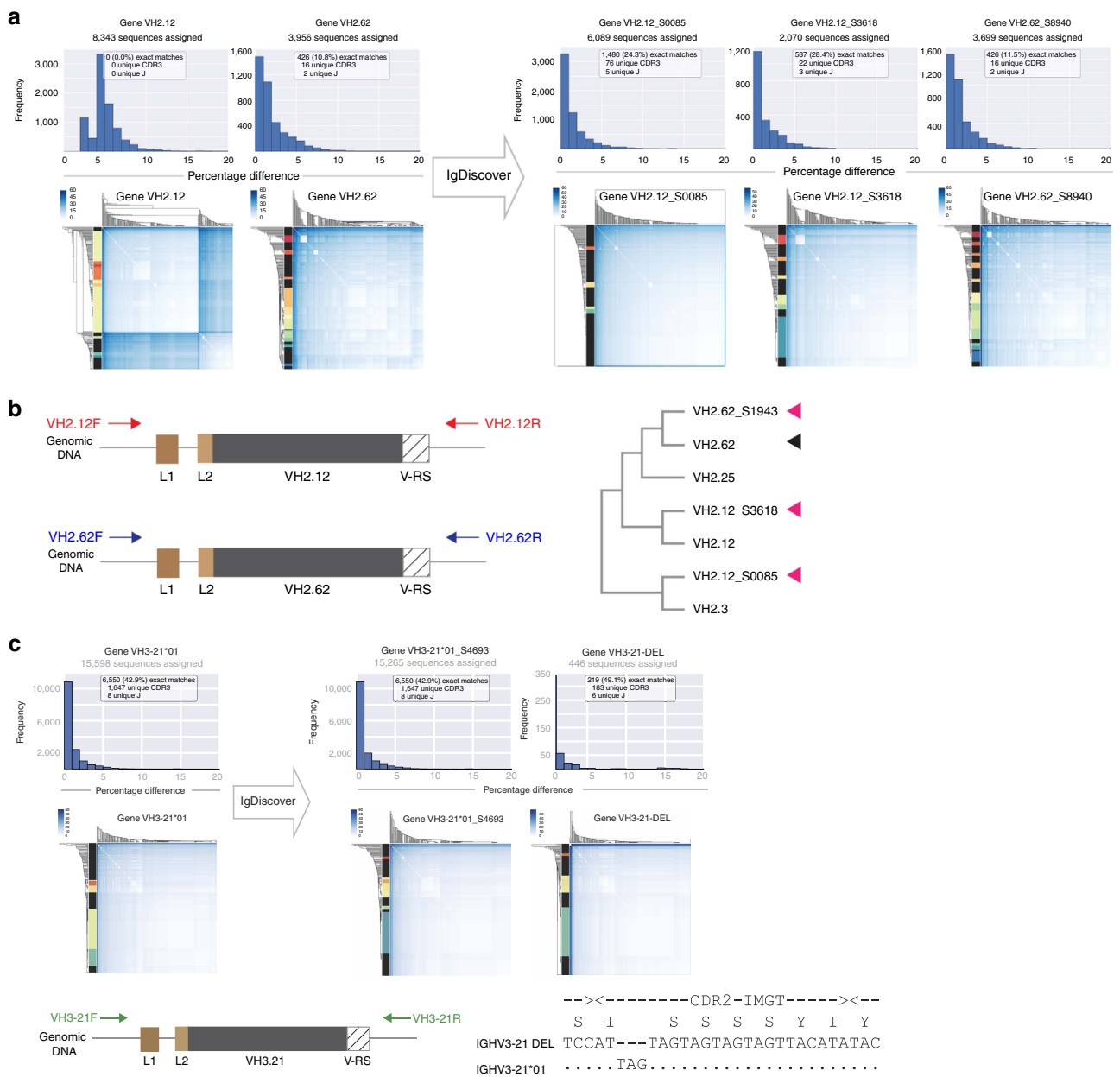

**Figure 7 | Genomic validation of novel rhesus and human alleles.** (**a**) Windowed cluster plots (upper left histograms) and Linkage cluster plots (lower left panels) of rhesus macaque F124 VH sequences assigned to starting database sequences VH2.12 and VH2.62, respectively. The upper left histograms and lower left linkage cluster plots show that starting database gene VH2.12 is not expressed in the F124 library. In contrast, VH2.62 is expressed, as indicated by the presence of 426 exact matches that use 16 unique CDR3s. Three novel VH2 family alleles found by IgDiscover, VH2.12_S0085, VH2.12_S3618 and VH2.62_S8940, are shown in the upper right histograms and lower right linkage cluster plots. (**b**) Position of primers used to amplify novel VH2 family alleles. The primers were designed to encompass the genomic sequence of previously validated VH2 family genes, VH2.12 and VH2.62 (primer sequences in Supplementary Table 1) resulting in identification of VH2.12_S0085, VH2.12_S3618 and VH2.62_S8940 in the genomic DNA. The right lower panel shows a phylogenetic tree indicating the relationship between the previously published VH2 family sequences and the four VH2 family alleles expressed in rhesus macaque F124. The three novel alleles and one previously known allele identified are indicated with pink arrows and black arrow, respectively. (**c**) Left panels illustrate the initial assignment of sequences from the H1 human library to VH3-21*01. The centre four panels demonstrate IgDiscover identification of two distinct sequences within the initial assigned group that are used in multiple independent V, diversity and J rearrangements. The right panels show the position of genomic primers used for genomic validation and the resulting identification of an unrearranged genomic sequence corresponding to a novel IGHV3-21 allele that differs from IGHV3-21*01 by exactly three nucleotides.

throughput antibody cloning techniques[24]. Furthermore, the upstream sequence information produced by IgDiscover has the potential to aid in the accurate assignment of antibodies, in particular those that have undergone significant levels of mutation in their V segments.

The functionality of the IgBLAST module, used within IgDiscover for gene assignment, is not restricted to heavy and light chain V gene analysis, as T-cell receptor V genes can also be processed by IgBLAST[13]. In all cases, the approach taken in the current study: assignment to a starting database,

**Table 1 | Genomic validation of IgDiscover identified germline sequences.**

| Genomic clone | Accession no. | VH allele | Accession no. | Allele status | Previous designation | PCR primer encompassing | Database source |
|---|---|---|---|---|---|---|---|
| 14626254 | KU593272 | VH6_B* | KU593026 | Known | VH6.1 | VH6.1 | Sundling et al.[31] |
| 14626253 | KU593299 | VH6_A | KU593104 | Novel | | VH6.1 | |
| 15485012 | KU593293 | VH4_1Q | KU592908 | Novel | | VH4.57 | |
| 15484988 | KU593292 | VH4_2T | KU592919 | Novel | | VH4.57 | |
| 15268538 | KU593305 | VH4_5K | KU592932 | Known | VH4.37 | VH4.37 | Sundling et al.[31] |
| 15484883 | KU593297 | VH4_2O | KU593314 | Novel | | VH4.34 | |
| 15484976 | KU593295 | VH4_3A | KU592922 | Novel | | VH4.34 | |
| 15485000 | KU593296 | VH4_5I | KU593154 | Novel | | VH4.34 | |
| 15484928 | KU593312 | VH4_2Q | KU592917 | Novel | | VH4.28 | |
| 15484880 | KU593310 | VH4_3C | KU593020 | Novel | | VH4.28 | |
| 15484904 | KU593311 | VH4_4K | KU593088 | Novel | | VH4.28 | |
| 15268571 | KU593306 | VH4_2L | KU592916 | Novel | | VH4.26 | |
| 15268574 | KU593307 | VH4_4I | KU592929 | Novel | | VH4.26 | |
| 15268577 | KU593308 | VH4_4Y | KU593150 | Novel | | VH4.26 | |
| 15268535 | KU593290 | VH4_1P | KU592907 | Novel | | VH4.22 | |
| 15268541 | KU593291 | VH4_2C | KU592911 | Novel | | VH4.22 | |
| 15692045 | KU593313 | VH4_3G | KU592926 | Known | VH4.11 | VH4.11 | Sundling et al.[31] |
| 14831513 | KU593284 | VH4_1C | KU592903 | Novel | | VH4.11 | |
| 14768083 | KU593282 | VH4_2F | KU592913 | Novel | | VH4.11 | |
| 14768075 | KU593281 | VH4_2U | KU592920 | Novel | | VH4.11 | |
| 15484898 | KU593294 | VH3_2N | KU592890 | Novel | | VH3.63 | |
| 14768082 | KU593277 | VH3_2F | KU592889 | Novel | | VH3.6 | |
| 14768090 | KU593303 | VH3_3J | KU592893 | Novel | | VH3.6 | |
| 14980506 | KU593285 | VH3_1C | KU592882 | Known | VH3.58 | VH3.58 | Sundling et al.[31] |
| 14768105 | KU593276 | VH3_3U | KU592894 | Known | VH3.58 | VH3.5 | Sundling et al.[31] |
| 14768097 | KU593275 | VH3_3O | KU592979 | Novel | | VH3.5 | |
| 16974832 | KU593300 | VH3_2R | KU593121 | Novel | | VH3.44 | |
| 14768109 | KU593279 | VH3_1W | KU592885 | Known | VH3.24 | VH3.24 | Sundling et al.[31] |
| 15268547 | KU593289 | VH3_3Z | KU592896 | Novel | | VH3.24 | |
| 14768117 | KU593280 | VH3_3B | KU592977 | Novel | | VH3.17 | |
| 14831517 | KU593283 | VH3_1G | KU592904 | Novel | | VH3.15 | |
| 14768100 | KU593278 | VH3_2X | KU592892 | Novel | | VH3.15 | |
| 14768116 | KU593304 | VH3_4E | KU593315 | Novel | | VH3.15 | |
| 14980509 | KU593286 | VH3_4L | KU593067 | Novel | | VH3.13 | |
| 15268532 | KU593288 | VH3_1V | KU592884 | Novel | | VH3.10 | |
| 14768099 | KU593302 | VH2_1A | KU592878 | Novel | | VH2.25 | |
| 15268559 | KU593287 | VH2_1M | KU592881 | Novel | | VH2.12 | |
| 14768115 | KU593274 | VH2_1S | KU593176 | Novel | | VH2.12 | |
| 15484937 | KU593309 | VH1_1S* | KU592996 | Known | | VH1.53 | Sundling et al.[31] |
| 15484916 | KU593298 | VH1_2B | KU592875 | Known | | VH1.23 | Sundling et al.[31] |
| 14768093 | KU593301 | VH1_1L | KU592947 | Novel | | VH1.16 | |
| 14768085 | KU593273 | VH1_1V | KU593032 | Novel | | VH1.16 | |

cluster identification, and application of a germline filter, will be applicable, with only minor adjustments necessary of IgDiscover for it to become functional as a pan Ig and T-cell receptor germline discovery tool. Antibody repertoire analysis is increasingly seen as a vital research tool that facilitates precise examination of B cell biology for a broad range of questions[25]. Increasing numbers of computational tools are being developed to facilitate accurate and rapid analysis of NGS data[26,27]. With the ability to now produce individualized germline databases de novo, IgDiscover both augments current techniques and, importantly, finally enables the extension of antibody repertoire analysis to multiple additional species.

## Methods

**Animals and ethics statement.** Previously described rhesus macaques (*Macaca mulatta*) of Chinese origin[28] were sampled in the present study. The animals were housed at the AAALAC accredited Astrid Fagraeus Laboratory at Karolinska Institutet. Housing and care procedures were in compliance with the guidelines of the Swedish Board of Agriculture. The facility has been assigned an Animal Welfare Assurance number by the Office of Laboratory Animal Welfare at the National Institute of Health. The Local Ethical Committee on Animal Experiments (Stockholms Norra Djurförsöksetiska Nämnd) (ethical permit number N85/09 and

N32/12) approved all procedures. Before inclusion in the study, all animals were tested and confirmed negative for simian immunodeficiency virus, simian T-cell lymphotropic virus, herpes simian B virus and simian retrovirus type D. Further samples were obtained from rhesus macaques (*M. mulatta*) of Indian origin kept at the Deutsches Primatenzentrum GmbH (DPZ), Leibniz-Institute for Primate Research in Goettingen under standard conditions complying with §§7–9 of the German Animal Welfare Act, which strictly adheres to the European Union guidelines (EU directive 2010/63/ EU) on the use of non-human primates for biomedical research. Experienced veterinarians and animal caretakers observed the monkeys constantly. Blood collection from the animals was approved by an external ethics committee of the Lower Saxony State Office for Consumer Protection and Food Safety with the project licence 33.9-42502-05-10A04 and performed under anesthesia with 10 mg ketamine per kg body weight. Animals were tested negative for antibodies against those mentioned above except for herpes B virus, which had not been analysed for. The human samples were donated following informed consent by healthy human blood donors and obtained from the Karolinska University Hospital blood bank and the mouse samples were obtained from C57BL/6J mouse and two Balb/c mice purchased from the Jackson laboratory and maintained in the animal research facility at the Department of Microbiology, Tumor and Cell Biology, Karolinska Institutet in accordance with the local Ethical Committee on Animal Experiments (Stockholms Norra djurförsöksetiska nämnd) approval.

**Cell isolation and RNA and genomic DNA extraction.** Peripheral blood lymphocytes from non-immunized animals and from the human sample

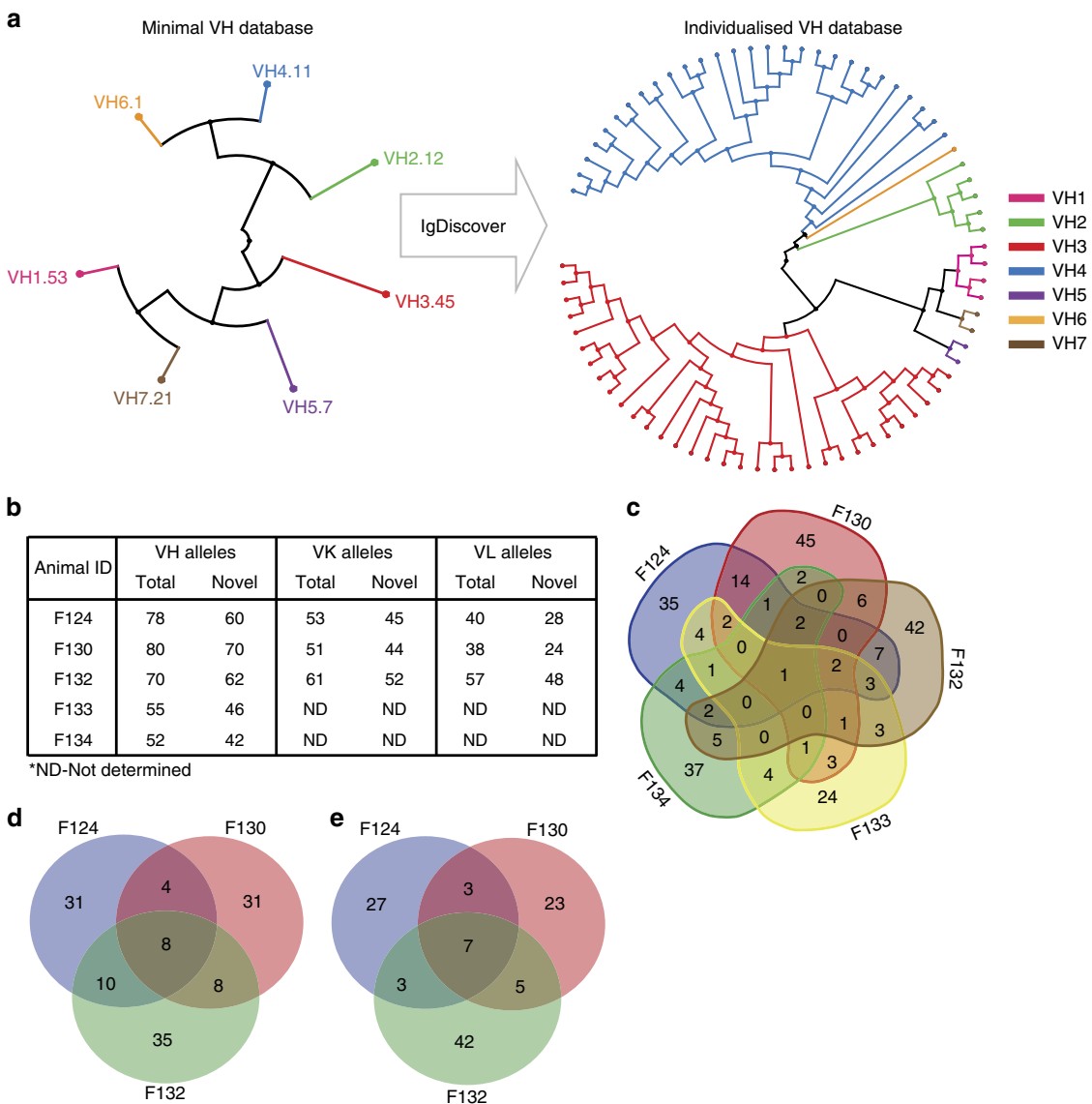

**Figure 8 | Definition of individualized V gene germline repertoires in the Chinese-origin macaques by IgDiscover. (a)** The left phylogenetic tree shows the relationship between the seven VH sequences that constitute the minimum VH database. The right phylogenetic tree illustrates IgDiscover identifying a repertoire of 70 VH alleles in rhesus F132 using a seven sequence minimal VH set as the starting database. The seven individual VH families are colour coded: VH1, pink; VH2, green; VH3, red; VH4, blue; VH5, purple; VH6, orange; and VH7, brown. **(b)** Total number of alleles and the number of novel VH alleles in each of the five Chinese-origin rhesus macaques analysed in addition to the V kappa and V lambda sequences from three of this group. **(c)** Venn diagram showing the comparison of VH allelic diversity within the group of five rhesus macaques. **(d)** Venn diagram showing the comparison of VK allelic diversity within the group of three rhesus macaques. **(e)** Venn diagram showing the comparison of VL allelic diversity within the group of three rhesus macaques.

$(2–5 \times 10^6$ cells) were isolated by density-gradient centrifugation with Ficoll-Hypaque (GE Healthcare). Total RNA and genomic DNA were isolated using the RNeasy Kit (Qiagen) and QIAamp Mini DNA Kit (Qiagen), respectively.

**5′ rapid amplification of cDNA ends.** Total RNA (400 ng) was used in a 5′RACE cDNA synthesis reaction. The first-strand cDNA synthesis was performed at 42 °C for 60 min using an oligo dT primer and Superscript II reverse transcriptase (Thermofisher), followed by a template switching step using a 5′ template switch oligonucleotide RACE primer SM_RACE1 at 42 °C for another 60 min. In all, 10% of the resulting cDNA was amplified using the Kapa HiFi Hotstart ReadyMix system (Kapa Biosystems) with the primers F_Universal and IgM_RevRhesus, IgM_RevMouse or IgM_RevHuman, respectively. All primer sequences are provided in Supplementary Table 1. The resultant bands of approximately 550 bp were gel purified using the Qiagen Gel Purification Kit, for subsequent NGS library production.

**NGS library production and sequencing.** A total of 200 ng of each gel-purified RACE-amplified IgM, IgK, IgL or IgG product was used in the production of an NGS ready library. Indexed adapters from the Trueseq Nano Kit (Illumina) were ligated to the RACE product, according to the manufacturer's instructions, and the resulting library was validated and quantified. The various individually indexed libraries were sequenced on the Illumina MiSeq using the Illumina Version 3, $(2 \times 300$ bp) Sequencing Kit. A total of 15% PhiX174 DNA was included as a control and as a means of generating diversity within the MiSeq flow cell. IgM libraries from five Chinese- and four Indian-origin rhesus macaques, in addition to one human and one mouse, were used in this current analysis. In addition, three Chinese-origin rhesus macaque libraries were used to produce VK and VL libraries for light chain germline analysis. The merged library size for each as IgM Library/Merged sequences is as follows: Human IgM/173595, Rhesus F124 Chinese IgM/492935, Rhesus F130 Chinese IgM/510229, Rhesus F132 Chinese IgM/631232, Rhesus F133 Chinese IgM/379765, Rhesus F134 Chinese IgM/338373, Rhesus 2635 Indian IgM/611431, Rhesus 2636 Indian IgM/505578,

**Table 2 | IgDiscover output of 5′UTR and Leader sequence from VH1 family alleles.**

| VH allele | Leader + 5′UTR |
|---|---|
| VH1_1A | GCATCACAAAACAACCACATCCCTCCTCTAAAGAAGCCCCTGGGAACACAGCTCATCACCATGGACTGTACCTGGAGGCTCCTCTTTGTGGTGGCAGCAGCTACAGGTGCCAAGTCC |
| VH1_1C | GATCACATAACAACCACATCCCTCCTCTAAAGAAGCCCCTGGGAACACAGCTCATCACCATGGACTGGACCTGGAGGCTCCTCTTTGTGGTGGCAGCAGCTACAGGTGCCAAGTCC |
| VH1_1D | GATCACATAACAACCACATCCCTCCTCTAAAGAAGCCCCTGGGAACACAGCTCATCACCATGGACTTGACCTGGAGGCTCCTCTTTGTGGTGGCAGCAGCTACAGGTGCCCAGTCC |
| VH1_1E | GCATCACATAACAACCACATCCCTCCTCTAAAGAAGCCCCTGGGAACACAGCTCATCACCATGGACTTGACCTGGAGGCTCCTCTTTGTGGTGGCAGCAGCTACAGGTGCCCAGTCC |
| VH1_1F | ATCGCCCAGCAACCACATCCCTTCTCTACAGAAGCCCCTGAGAGGAAAGCTCTTCACCATGGACTGGACCTGGATGGTCTTCTGCTTGCTGGCAGTAGCTCCAGGGGCCCACTCC |
| VH1_1G | GACTCACTCAACAACCATATTCCTCCTCTGGAGAAAACCCTGGAACTGCAGCTCCTCACCATGGACTGGACCTGGAGGATCCTCTTCCTTGTGGCAGCAGCTACAGGTGCCCAGTCT |
| VH1_1H | CACTCAACAACCACATCCCTCCTCTGGAGAAAACCCTGGAACCGCAGCTCCTCACGATGGACTGGACCTGGAGGATTCTCTTCCTTGTGGCAGCAGCTACAGGCGCCCAGTCT |
| VH1_1I | GACTCAACAACTGCATCCAACCTCAAGAGAAGCCCCTGAGAGCACAGTTTCTCACCATGGACTTGACCTGGAAGATCCTCCTCTTGGTGACAGCAGCCACAGGTGCCCACTCC |
| VH1_1J | AGCATCACACAACAACCACATCCCTCCCCTACAGAAGCCCCAGAGCACAGCACCTCACCATGGACTGGACATGGAGGATCCTCCTCTTGGTGGCAGCAGCTACAGGTGCCCACTCC |
| VH1_1K | ATCACACAGCAACCACATCCCTCCCCTACAGAAGCCCCAGAGCACAGCACCTCACCATGGACTGGACATGGAGACTCCTCCTCTTGGTGGCAGCAGCTACAGGTGCCCACTCC |
| VH1_1L | GAGCATCACACAACAACCACATCCCTCCCCTACAGAAGCCCCAGAGCACAGCACCTCACCATGGACTGGACCTGGAGGCTCCTCCTCTTGGTGGCAGCAGCTACAGGTGCCCACTCC |
| VH1_1M | GAACATCACACAACAACCACATCCCTCCCCTACAGAAGCCCCAGAGCACAGCACCTCACCATGGACTGGACATGGAGGCTCCTCCTCTTGGTGGCAGCAGCTACAGGCGCCCACTCC |
| VH1_1N | GAGCATCACACAACCACCACATCCCTCCCCTACAGAAGCCCCAGAGCACAGCACCTCACCATGGACTGGACCTGGAGGCTCCTCCTCTTGGTGGCAGCAGCTACAGGCGCCCACTCC |
| VH1_1O | GAGCATCACACAACAACCACATCCCCTACAGACGCCCCCAGAGCACAGCACCTCACCATGGACTGGACCTGGAGGCTCCTCCTCTTGGTGGCAGCAGCTACAGGCGCCCACTCT |
| VH1_1T | GAGCATCACACAACAACCACATCCCTCCCCTACAGAAGCCCCAGAGCACAGCACCTCACCATGGACTGGACCTGGAGGATCCTCCTCTTGGTGGCAGCAGCTACAGGCGCCCACTCT |
| VH1_1V | GACTCAACAACTGCATCCAACCTCAAGAGAAGCCCCTGAGAGCACAGTTTCTCACCATGGACTTGACCTGGAAGATCCTCCTCTTGGTGACAGCAGCCACAGGTGCCCACTCC |
| VH1_1W | ACTCAACAACTGCATCCAACCTCAAGAGAAGCCCCTGAGAGCACAGTTTCTCACCATGGACTTGACCTGGAAGATCCTCCTCTTGGTGACAGCAGCCACAGGTGCCCACTCC |

Rhesus 2514 Indian IgM/679119, Rhesus 5200 Indian IgM/367104, Human H1/1142690, Human H2/1088767, and Human H3/1003720.

**Technical requirements.** Ig sequences obtained through NGS are subject to the technical limitations of the current methodologies, namely, sequence length and average sequence error rate. In our case, the Illumina MiSeq platform was chosen as it can produce sequence reads of $2 \times 300$ bases and is known to have one of the lowest error rates, approximately 1.2% (refs 29,30), of current NGS sequencing technologies. Because IgDiscover works on the basis of identifying consensus sequences based on clusters assigned to an initial database, the process requires approximately 400,000 initial paired sequences in order to identify a full expressed germline repertoire from an individual. Libraries with lower numbers of sequences will result in specific VH germline detection; however, there may be insufficient numbers of individual sequences for all the VH alleles to pass the germline filter. In addition, as exact sequence identity is used as an integral part of the process, the process functions more effectively in sequence runs with lower sequencing error rates. IgDiscover works efficiently with IgM libraries. IgG libraries contain lower proportions of germline sequences and are not processed efficiently by the program.

**Targeted genomic sequencing.** Germline rhesus VH sequences from the database published in Sundling et al.[31] were screened against the Indian and Chinese rhesus genomic reference sequences (MGSC Merged1.0/rheM ac2 and BGI CR_1.0/rheMac3, respectively) and to an additional reference sequence (Mmul-8.0.1 reference assembly, Annotation Release 102) from an extended reference genomic sequence produced by Zimin et al.[32] from the same Indian-origin rhesus used to produce the MGSC Merged1.0 reference. This resulted in the identification of genomic segments that showed >99% identity to the database germline sequences and allowed us to design PCR primers, shown in Supplementary Table 1, to amplify genomic DNA that encompassed these VH genes. PCR amplification was performed using each set of primers. Primer sequences were arranged so that the primers were targeted to genomic sequence external to the VH exons in question in order to avoid amplification of rearranged DNA.

**Preprocessing.** IgDiscover first merges paired-end reads with PEAR[33] and optionally removes primer sequences with cutadapt[34]. Sequences of at least 300 bp are converted from FASTQ to FASTA and passed to VSEARCH's derep-fulllength command to remove duplicates. The resulting sequences are used as input for all discovery iterations.

**Gene and CDR3 assignment.** IgBLAST is run on the preprocessed sequences at the beginning of each iteration with default algorithm parameters, except that a custom database is supplied. IgBLAST output is parsed and stored as a tab-separated value file for easier processing[35]. Following an idea by D'Angelo et al.[36], CDR3 sequences of heavy and light chains are detected using a regular expression at the amino acid level.

**Quality filtering.** After gene assignment, only sequences passing a quality filter are kept: Both V and J genes must be assigned, no stop codon must occur and the E-value of the hit must be at most $10^{-3}$. In addition, we require that the detected VH gene region in the query sequence must cover at least 90% of the reference VH sequence, and the JH gene must be covered by at least 60% in the same way, which reduces the number of spurious hits.

**Linkage cluster analysis.** To reduce computation time, not all sequences assigned to a database reference allele are clustered with the UPGMA algorithm (average linkage) but only a random subsample of 1,000 sequences. If the starting database is very small, subsampling may lead to genes that are expressed at low levels not being detected as they represent a too small fraction of the subsample. With larger starting databases, concordance is typically very high. As distance function, we use Levenshtein distance (insertion, deletion and substitution operations count as one difference each). Hierarchical clustering results in a tree structure (shown as a dendrogram in the clusterplots, such as Fig. 3f). To detect appropriate subclusters, a heuristic is used that finds those inner nodes of the tree having two subtrees that both have a size of at least five. If the ration between the smallest distance in one subtree (A) to the largest distance in the other subtree is <0.8, subtree A is detected as a subcluster. This ensures that a subcluster is sufficiently dissimilar from its neighbouring sequences.

**Germline filter.** Candidate sequences are filtered according to the following criteria. The consensus sequences must contain neither ambiguous ('N') bases nor stop codons and must have been computed from a cluster of at least 100 sequences. To test for independent rearrangements, all exact occurrences of the sequence in the full data set are inspected and must be associated with at least ten different CDR3 sequences and with at least three unique J genes. If two candidate sequences differ by up to two bases from each other, the one associated with the lowest number of unique HCDR3 is removed. When comparing sequences, small length differences at the 5' and 3' ends are not counted as errors, but the longer sequence is preferentially kept. Filtering criteria for the pregermline filter are identical except that it does not apply a minimum cluster size and only at least two unique HCDR3 sequences and at least two unique J genes are required. If a consensus sequence is encountered that is identical to one in the initial database, this is considered to be sufficient evidence and the filtering criteria are not applied. This 'whitelisting' makes it possible to retain sequences occurring at low expression levels that would otherwise not pass filters.

**Upstream sequence detection.** For primer design, it is useful to know the sequence upstream of each VH allele. This consists of the 5'UTR and part of the leader sequence. IgDiscover can detect the upstream sequence. For each allele, it takes all the processed sequences assigned to that allele that also have a low error rate in the VH allele match. A consensus is then computed from all sequences that are at least as long as the tenth longest one and this is then output as the upstream sequence.

**Implementation.** IgDiscover consists of a set of command-line scripts written in Python and a workflow implemented in Snakemake[37] that automatically runs the scripts and other tools in the correct order and with the necessary parameters. SciPy (http://www.scipy.org/) is used for its hierarchical clustering functionality, seaborn[38] and matplotlib[39] for plotting and pandas and numpy for working with tabular data. Cython[40] was used to speed up some algorithms.

VSEARCH[41] was used as a quality filter. Consensus sequences are found by computing a multiple alignment with MUSCLE[42] and reporting the base at each position that occurs with a 60% majority. IgDiscover makes intermediate results available as text files and automatically creates plots such as those shown in Figs 2 and 4. Additional scripts are included that allow comparing and combining VH allele databases.

**Runtime.** IgDiscover running time depends, among other factors, on the number of reads that can be merged, the number of available CPU cores and the number of requested iterations. We measured runtime on a human data set with 1.4 million reads of good quality on an Intel Xeon E5-2660 with 16 cores. As the human database is already nearly complete, a single iteration is sufficient. In this best-case scenario, IgDiscover needs 95 min to discover a fully personalized V gene database. Preprocessing and the discovery process itself (including cluster analysis and consensus building) account for <10% of this time, whereas the remaining >90% are spent on running IgBLAST for obtaining gene assignments. In practice, IgDiscover runs three iterations by default to cover those cases where the input database is more distant to the analysed sample. It also executes IgBLAST one additional time so that a gene usage profile based on the final personalized database can be computed. Runtime in this case, assuming again a 16-core CPU, is ca. 5 h.

**Software availability.** IgDiscover is available as Open Source software under the MIT license from: http://bitbucket.org/igdiscover/igdiscover

**Data availability.** Sequencing data supporting the findings of this study have been deposited in the European Nucleotide Archive under accession PRJEB15295. Sequences identified in this study are available from Genbank database under the following accession numbers: KU592872 - KU592940 (Rhesus F124 Chinese IgM); KU592941 - KU593026 (Rhesus F130 Chinese IgM); KU593027 - KU593106 (Rhesus F132 Chinese IgM); KU593107 - KU593164 (Rhesus F133 Chinese IgM); KU593165 - KU593218 (Rhesus F134 Chinese IgM); KU593219 - KU593271, Rhesus 2635 Indian IgM; KX055207 - KX055255 (Rhesus F124 Chinese IgK); KX055256 - KX055311(Rhesus F130 Chinese IgK); KX055312 - KX055349 (Rhesus F132 Chinese IgK); KY199293 - KY199335 (Rhesus F124 Chinese IgL); KY199336 - KY199377 (Rhesus F130 Chinese IgL); KY199378 - KY199422 (Rhesus F132 Chinese IgL); KY198750 - KY198943 (Human VH sequences from H1, H2 and H3 libraries); KY198944 - KY199292 (Mouse VH sequences from M1, M2 and M3 libraries); KU593272 - KU593313 (Rhesus Genomic validation); KY110713 - KY110714 (Human Genomic validation).

The authors declare that all other data supporting the findings of this study are available within the article and its Supplementary Information files or from the corresponding authors upon request.

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

## Acknowledgements

This work was funded by an NIH P01 grant AI104722 (HIVRAD) and a grant from the International AIDS Vaccine Initiative (IAVI) to G.K.H. G.E.P. is funded through a Karolinska Institutet KID grant, and M.M. is supported by a grant from the Knut and Alice Wallenberg Foundation to the Wallenberg Advanced Bioinformatics Infrastructure. We thank Rolf Ohlsson and Anita Göndör at the Department of Microbiology, Tumor and Cell Biology, Karolinska Institutet for generously allowing us to use their Illumina MiSeq instrument. We thank Paola Martinez for kind assistance with sample preparation and Pär Engström and Christopher Sundling for providing helpful comments on the manuscript.

## Author contributions

M.M.C. and G.E.P. conceived the study, prepared the libraries, developed the algorithms, analysed the data and wrote the paper. M.M. conceived the study, developed the algorithms and wrote the code for the program, analysed the data and wrote the paper. N.S. assisted with the sequencing experiments. C.S.-H. provided some of the experimental animals used in the study. N.V.B. performed the genomic cloning. M.A.A.P. conceived the study. and G.B.K.H. conceived the study, analysed the data and wrote the paper.

## Additional information

**Competing financial interests:** The authors declare no competing financial interests.

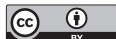

