## [Peer Review File · Nature Communications]

Reviewers' comments:

Reviewer #1 (Remarks to the Author):

1. In the introduction (line 67, page 3) the authors claim to validate the IgDiscover by re-discovering human VH alleles starting from an artificially reduced database. The validation of IgDiscover with human VH alleles is illustrated in Figure 2 and validation with mouse VH alleles is shown in Figure 3. The ultimate claim is that IgDiscover produces individualized germline databases de novo (line 468, line 494). However, the authors do not show a complete reconstruction of already existing and validated human (55 VH genes Figure 1b, and Figure 2a-f) or mouse VH genes (VH 218, Figure 1b, and Figure 3b).

It seems that the screening database of human alleles is reduced to the only VH1 family, and to only 8 alleles. The screening database of mouse alleles seems to have been reduced to the only VH1 family as well.

Therefore, the authors should demonstrate they are able to rediscover the entire database of human and mouse VH genes.

2. In Figure 2a, a candidate novel allele (IGHV1-24*02) from the consensus of sequences is recovered different from the deleted allele IGHV1-24*01. The authors do not show the recovered allele in the phylogenetic relation in the left panel, neither do they give an explanation for the different output of the algorithm from the removed allele, making it more difficult to assess the validity of the tool. It is stated that the IgDiscover recovered allele is 1-20% identical to IGHV1-24*01, yet this range does not explain why the original allele could not be recovered. Also, where an exact similarity can be reported, the range is indicated. The authors should be more clear regarding the allele similarity ranges they report in Figure 2 and how do these support their conclusions.

3. Related, to point 1. - the mouse validation shown in Figure 3b, indicates that IgDiscover recovers 36 final germline sequences that are 100% identical to IMGT. It seems that only IGHV1 gene alleles were analyzed. "The proportion of validated germline sequences reached 100% after germline filtering," yet the authors fail to make explicit this proportion, or the proportion of validated germline sequences that IgDiscover recovers against the complete known database (349 total VH alleles, Figure 1b, line 109).

4. The authors apply IgDiscover to germline VK and VL light chain sequences (line 381, 446) and report increasing of the VL database by 21 allelic sequences, yet they don't show results regarding the VL recovery as for VK (Figure8d-e), number of animals, etc.

5. Only merged sequences are reported (Table in page 26, and line 576), yet it would be helpful to the reader to know what range of pre-processed (or unique) sequences does IgDiscover take as input in order to assign germline sequences. Also, the number of merged sequences or pre-processed sequences are not shown for light chains. On an additional note, it would be useful to the reader to know how read-quality affects the germline detection of IgDiscover. This information should be included in an additional table.

6. It is shown that in the case of rhesus macaques (Figure 5e) the IgDiscovery analysis repeat would output 2 sequences specific to this repetition, and would find 79 out of 80 germline sequences from the previous IgDiscovery analysis. The authors could clarify if and how the iteration number (Figure 4c) affects the final size of the resulting database.

7. Quality control is mentioned in read pre-processing (line 222) but no further clarification on what quality thresholds were set to select for reads.

8. The authors report that a random subsample of 1000 sequences are clustered for Linkage cluster analysis. It would be useful to the reader to know how the sampling affects the discovery of germline sequences.

9. Number reporting inconsistency

- The authors report 74 novel VH alleles identified for rhesus F132 (lines 369-370, Figure 8a). In Figure 8b, 71 novel VH alleles for F132 are reported instead.

- Figure 8d-e reports incongruent numbers between table and venn diagram (panel e).

F130 shows 47 total VK alleles in venn diagram while 49 are shown in Figure 8d.

F124 shows 36 total VK alleles in venn diagram while 38 are shown in Figure 8d.

10. Figure 1b states that the Rhesus monkey has 106* alleles/genes. The authors claim in figure 5 that they used all of these alleles for IgDiscover (5A), but then only 80 alleles were returned. There does not appear to be an explicit statement of the ratio of VH genes to alleles, like what is done human and mouse. It would be important to know as to why only 80 alleles were returned, and if across all of the starting databases these same alleles were excluded. This would show some kind of systematic bias that would be important to mention.

11. The authors should report how fast IgDiscover performs as a software platform. For example, the relationship of how fast de novo V-gene germlines are discovered starting from X number of NGS reads and X number of starting V-gene database should be included. This will help for future users to calibrate their datasets for implementation of the method.

12. Lines 120-121, I suggest the authors refrain from describing '5' RACE as an "unbiased" amplification. Several papers have shown that substantial artifacts can occur during 5' RACE, (see: Cocquet et al., Genomics, 2006 and Best et al., Sci Reports, 2015). I understand that the authors probably mean that 5' RACE helps to avoid some of the biases and mispriming caused by multiplex-PCR, thus they should be more explicit to state this rather than simply calling 5' RACE unbiased.

13. Throughout the manuscript the authors should be more specific about which mouse strain they are using or referring to. For example, only the C57BL6 mouse strain has a fully sequenced and annotated genome and immunoglobulin locus. This not however the case for Balb/c.

Reviewer #2 (Remarks to the Author):

This paper describes a new computational method to infer germline immune gene segments, based on next gen immune repertoire sequencing. The paper also suggests a message for personalized databases of immune repertoire genes.

The topic is timely, since this field of immune repertoire sequencing is exploding, both in humans and experimental animals, but the pipelines to analyze these sequence sets, and the databases of gene

segments with which to align sequences is incomplete or missing. The method described here offers a way to fill that gap until genomic gene segments are fully defined, and also offers a personalized way to analyze the repertoires of individual people or animals with polymorphisms.

The paper is very dense. The iterative nature of the computational work and the lab validation is important, but complex to follow. Given it is essentially a methods paper, I don't have any specific suggestions to decomplexify the paper, since aficionados who will try to use this method will want to see the detail.

The data is well described, and the methods for both computation and lab seem state of the art. All details needed to reproduce the lab work are provided, and the code has been made accessible.

No doubt there is some fuzziness in the assignment of gene segment numbers or sequences in this paper, but that is the nature of the work at this point, and is ok

REVIEWERS' COMMENTS:

Reviewer #1 (Remarks to the Author):

I thank the authors for the thorough and thoughtful responses to my questions. They have done an excellent job in revising the manuscript and improving the few areas of concern.

Reviewer #2 (Remarks to the Author):

The response to the previous reviewer queries appears thorough and improves the manuscript, especially the reconstruction of nearly complete gene sets.

Please find below our point-by-point response to the review of manuscript NCOMMS-16-09490.

The Reviewers' comments are pasted in below *in italics* with our response in normal font.

Reviewer #1:

1. In the introduction (line 67, page 3) the authors claim to validate the IgDiscover by re-discovering human VH alleles starting from an artificially reduced database. The validation of IgDiscover with human VH alleles is illustrated in Figure 2 and validation with mouse VH alleles is shown in Figure 3. The ultimate claim is that IgDiscover produces individualized germline databases de novo (line 468, line 494). However, the authors do not show a complete reconstruction of already existing and validated human (55 VH genes Figure 1b, and Figure 2a-f) or mouse VH genes (VH 218, Figure 1b, and Figure 3b).

It seems that the screening database of human alleles is reduced to the only VH1 family, and to only 8 alleles. The screening database of mouse alleles seems to have been reduced to the only VH1 family as well.

Therefore, the authors should demonstrate they are able to rediscover the entire database of human and mouse VH genes.

Response:

The IgDiscover V gene discovery process functions through the identification of unmutated naïve germline sequences present within the expressed antibody repertoire of an individual, either human or other species. Due to genotypic diversity, the repertoire of germline V gene sequences may differ significantly between individuals, and the utilization of a broad expression range of V sequences is necessary to produce an individualized repertoire.

To ensure that low and extremely low frequency V gene sequences are identifiable by IgDiscover we have now modified the program. It can now utilize just 5 individual germline sequences involved in independent rearrangements, a potential frequency 0.000005% of a library of 1 million sequences. This high sensitivity modification is detailed as 'whitelisting' and explained in the germline filter section of the materials and methods.

We agree with the Reviewer that a reconstruction of individualized repertoires of existing validated genes would be an excellent test of the IgDiscover process. To this end we have now used IgDiscover to create individualized VH repertoires from three human samples and, additionally, from three mouse samples.

Analysis of the three human libraries enabled the identification of sequences corresponding to each of the 55 previously defined functional human gene VH genes (designated functional [F] within the IMGT human reference database) – with the exception of just two genes: IGHV3-NL1*01, a sequence isolated from an individual from the Papua New Guinea Highlands, and IGHV3-72*01, a sequence that was originally isolated from genomic DNA in 1988 through

its ability to hybridize to a mouse VH sequence. That is, of the 52 functional human VH genes deposited in IMGT, 50 sequences (100% identical) were re-identified by applying IgDiscover to IgM libraries made from three individuals.

Berman, J.E et al, Content and organization of the human Ig VH locus: definition of three new VH families and linkage to the Ig CH locus EMBO J. 7 (3), 727-738 (1988)

Given the geographically specific origin of IGHV3-NL1*01 it is entirely expected that we do not see this sequence in our data, which were generated from 3 individuals with genetic backgrounds quite distant from the individual carrying IGHV3-NL1*01.

While IGHV3-72*01 is present within the human reference genome, no example of expressed antibodies with VH segments identical to the full length of IGHV3-72*01 are present within the current Genbank nucleotide database, indicating that the gene is either extremely rarely expressed in the population in general – or that it is a non-functional pseudogene.

Our results are entirely consistent with either scenario and since IgDiscover is now sensitive to extremely low level expression, we are confident that the gene is not expressed in any of the three human samples.

The functionality of genes within the reference database is currently an issue of some debate since many reference sequences were originally denoted as functional not because they have been found expressed in antibodies, but because they contained uninterrupted reading frames and recombination signal sequences. The modifications we have now made to IgDiscover enable the identification of genes with extremely low frequency expression and so we feel it can be a valuable tool to enable the validation of candidate functional sequences.

IgDiscover was also used to analyze libraries from two Balb/c mice and one C57BL6 mouse. While the current mouse V database is far from ideal as a comprehensive and validated resource the production of individualized repertoires for the three mice libraries has been extremely valuable.

Analysis of the two Balb/c libraries identified a total of 140 individual VH sequences present in the two individualized repertoires. Importantly, IgDiscover analysis revealed a high degree of overlap between the two individual Balb/c mice (88 sequences identical between mouse 1 and mouse 2).

The low degree of overlap between the germline repertoires of Balb/c and C57BL6 mouse, (10 out of 140 sequences present in the Balb/c mice are identical to sequences from the C57BL6 mouse) provides extremely strong confirmation of the recent study by Andrew Collins and colleagues that first revealed similar low levels of overlap.

Collins, A.M., Wang, Y., Roskin, K.M., Marquis, C.P. & Jackson, K.J. The mouse antibody heavy chain repertoire is germline-focused and highly variable between inbred strains. *Philosophical transactions of the Royal Society of London. Series B, Biological sciences* **370** (2015)

Unlike the case of the human VH repertoire where there is a general consensus on the total numbers of V genes in any one individual – and therefore a defined target number by which IgDiscover can be judged – the same, at the current point in time, cannot be said for mouse.

Nonetheless we are confident that IgDiscover enables us to tackle this issue for three reasons.

1. We demonstrate in this paper that IgDiscover can identify germline VH sequences that differ by up to 30% from the initial starting library. This huge range of potential sequence divergence means that the use of a more comprehensive starting database, such as the current IMGT mouse reference database, ensures that both known sequences that are identical to the starting database, and novel sequences that are up to 30% different from starting database sequences can be identifiable. Importantly, additional iterations of the gene discovery process, which is a customizable part of the protocol, enable us to extend this to identify novel sequences with even greater levels of divergence.

2. The IgDiscover process originally used 5'RACE libraries that were subsequently sequenced using the Illumina MiSeq instrument. We have now extended the process to enable the use of antibody libraries produced by additional amplification methods and sequenced on multiple platforms. Importantly we have developed a multiplex PCR approach as part of the IgDiscover individualized repertoire process for both human and mouse, a technically less demanding technique compared to 5'RACE and thus applicable to a greater number of research groups. As evidenced by the human repertoire data, the use of the IgDiscover human multiplex primer set enables the identification of all previously known expressed functional VH genes.

The mouse VH reference database is not validated to anywhere near the same level as the human database, so while we can be confident that IgDiscover can identify germline sequences present in an NGS sequence library, we cannot be 100% certain that the current mouse multiplex PCR set amplifies every expressed V gene. Likewise it is an open question whether a single set of PCR primers will enable the amplification of all expressed VH genes in all strains of mouse. The ultimate solution to these issues is the comprehensive characterization of the expressed germline repertoire of all strains of mice in use in research and the identification of the full range of VH sequences, both currently known and novel. That, of course, is a formidable undertaking that will no doubt require numerous independent studies by a number of different research teams and, while far beyond the scope of the current study, it is exactly the kind of task that IgDiscover can in future facilitate.

In response to the specific point about the number of sequences in our human and mouse screening database, we think we must have been a little unclear in our presentation of these data. The individual databases produced in our human and mouse studies contain at minimum 58 sequences, extending across multiple VH families. For the human examples of re-finding genes deleted from the starting database the VH1 family was chosen for the simple reason that it is easier to represent the result in readable form compared to the much larger VH families such as VH3 or VH4 where the large number of sequences would render the point difficult for the reader to see. The mouse VH1 set in the germline filter figure was chosen at random and we apologize if it was unclear to the reviewer that a full set of sequences can be produced by IgDiscover.

We have altered the manuscript to take these points into account as follows:

1. We have now included a full repertoire identified by IgDiscover for all three humans and all three mice as supplementary figures.
2. We have altered the Figure describing the germline filter (Figure 3) to use a human library and shown examples from multiple gene families.

2. In Figure 2a, a candidate novel allele (*IGHV1-24*02*) from the consensus of sequences is recovered different from the deleted allele *IGHV1-24*01*. The authors do not show the recovered allele in the phylogenetic relation in the left panel, neither do they give an explanation for the different output of the algorithm from the removed allele, making it more difficult to assess the validity of the tool. It is stated that the IgDiscover recovered allele is 1-20% identical to *IGHV1-24*01*, yet this range does not explain why the original allele could not be recovered. Also, where an exact similarity can be reported, the range is indicated. The authors should be more clear regarding the allele similarity ranges they report in Figure 2 and how do these support their conclusions.

Response:

We apologize that a typographical mistake made during the production of this figure has led to this misunderstanding and we thank the reviewer for pointing out this to us.

In the legend to the figure 2 and in the main text of the results section we point out that the original sequence we deleted, which was *IGHV1-24*01*, was recovered during this experiment. The correct label beneath this panel should therefore be *IGHV1-24*01* – since we point out that the consensus sequence identified was identical to *IGHV1-24*01*.

Incidentally there is no *IGHV1-24*02* allele in the IMGT reference database, hence the reason why it would not appear on the phylogenetic tree. Again we apologize for this mistake.

Additionally we have altered the text on page 9 to make it clearer that the consensus in this case was generated from all sequences assigned to *IGHV1-f01* that differed by less than 20% identity to this gene.

3. Related, to point 1. - the mouse validation shown in Figure 3b, indicates that IgDiscover recovers 36 final germline sequences that are 100% identical to IMGT. It seems that only *IGHV1* gene alleles were analyzed. "The proportion of validated germline sequences reached 100% after germline filtering," yet the authors fail to make explicit this proportion, or the proportion of validated germline sequences that IgDiscover recovers against the complete known database (349 total *VH* alleles, Figure 1b, line 109).

Response:

The purpose of this figure was to show the ability of the germline filter process to remove non-germline sequences. This is a critical aspect of IgDiscover since by eliminating false positives it enables the efficient generation of a stable number of output sequences when libraries of different sizes are analyzed. Without an efficient germline filtering process the cumulative effect of small numbers of false positives would overwhelm the output.

As we mentioned in our response to point 1, we have now changed this figure to show the results of germline filter being applied to a human library. This change allows more gene families to be represented in the figure, so we hope it is now clearer that we have applied the filter to all sequences identified, rather than simply those from one subgroup.

Additionally, as the human database is more comprehensive, it is more obvious that the sequences being removed by the germline filter are, in fact, non-germline, i.e. not 100% identical to sequences from the human reference database.

Regarding the final point above

“The proportion of validated germline sequences reached 100% after germline filtering,” yet the authors fail to make explicit this proportion, or the proportion of validated germline sequences that IgDiscover recovers against the complete known database (349 total VH alleles, Figure 1b, line 109).”

The original mouse library in the previous version of Figure 3 was utilized for the purpose of testing whether the germline filter process would work across multiple species rather than as a means of generating a full repertoire in mouse. In the current revision process, we have subsequently produced three libraries from mice, as described above, of sufficient size to identify a full set of expressed VH germline sequences from those three animals.

Nevertheless, in terms of the question regarding the proportion of recovered sequences compared to the complete known reference database, we feel that the mouse database is not at a level of completeness, (in contrast to the human database) that such a figure would provide adequate information about the individual animal in question.

While the complete reference database may contain several hundred alleles, this is the cumulative figure for the species, not the number present in an individual animal. In a single animal the number of alleles present and capable of being expressed will necessarily be significantly lower. Unless one can be certain about the number of functional VH genes present in an individual animal or human, it will be impossible to accurately determine whether the total number of sequences identified has adequately represented the expressed germline repertoire of that individual. Since we have a much better idea about the total number of genes to expect in human samples, we realize that the human library is more suitable for this figure.

We realize now that the numbers for genes/alleles in figure 1 may be misleading. Many of these reference sequences are not expressed (for example pseudogenic sequences and V gene ORFs that lack the necessary recombination signal sequences that would enable their use in expressed functional antibodies.)

We have therefore altered these numbers to restrict the sequences to those marked as functional (“|F|” in the IMGT header) – a change that should more closely reflect the numbers of sequences that could possibly be expressed. We should not expect to see expression of pseudogenic or ORF sequences and so the absence of these in our output is a plus point rather than a negative.

4. The authors apply IgDiscover to germline VK and VL light chain sequences (line 381, 446) and report increasing of the VL database by 21 allelic sequences, yet they don't show results regarding the VL recovery as for VK (Figure 8d-e), number of animals, etc.

Response:

We have now added performed the required additional experiment and analyses and added the data for the rhesus lambda gene discovery as shown in Figure 8b and 8e.

5. Only merged sequences are reported (Table in page 26, and line 576), yet it would be helpful to the reader to know what range of pre-processed (or unique) sequences does IgDiscover take as input in order to assign germline sequences. Also, the number of merged sequences or pre-processed sequences are not shown for light chains. On an additional note, it would be useful to the reader to know how read-quality affects the germline detection of IgDiscover. This information should be included in an additional table.

Response:

We agree with the Reviewer that this information would be helpful to the reader. IgDiscover can identify germline sequences from libraries of less than 100,000 merged sequences, however in that case it is efficient at finding only the highest expressed germline sequences and so we recommend higher numbers of sequences.

We have added this point to the manuscript on page 28.

Additionally, we have run a series of experiments to determine the effect of sequence error on the ability of IgDiscover to detect germline sequences. Because IgDiscover exact sequences in its germline discovery process increased error rates lead to a decrease in the number of germline genes that are identifiable rather than leading to a large increase in false positives. We have included this data as a supplementary table.

6. It is shown that in the case of rhesus macaques (Figure 5e) the IgDiscovery analysis repeat would output 2 sequences specific to this repetition, and would find 79 out of 80 germline sequences from the previous IgDiscovery analysis. The authors could clarify if and how the iteration number (Figure 4c) affects the final size of the resulting database.

Response:

We do clarify this point in a later figure. We show the effect of the iterative database replacement process in figure 5H. IgDiscover can use a variety of starting databases, from a full species specific database to a single gene. It replaces this database with a new database of candidate germline sequences identified. This set of germline sequences becomes increasingly closer to the full expressed repertoire of that individual by each iteration. At some point, however, no additional expressed sequences will be identifiable since each individual will only express a finite number of alleles. The current version of IgDiscover enables this point to be reached after three iterations for most starting databases and perhaps a single iteration when using a comprehensive database.

7. Quality control is mentioned in read pre-processing (line 222) but no further clarification on what quality thresholds were set to select for reads.

Response:

We have clarified the figure caption. 'Quality controlled' in this sentence refers only to the removal of sequences shorter than 300 bp. Using that term is a leftover from earlier versions of the pipeline in which we indeed had more elaborate filtering schemes, but since we lost too many usable sequences doing this, we now pass on nearly all sequences to IgBLAST, and perform almost most of the quality filtering after IgBLAST assignment. Details of this

filtering, including thresholds, are described in Materials and Methods. We now refer the reader to Materials and Methods.

8. The authors report that a random subsample of 1000 sequences are clustered for Linkage cluster analysis. It would be useful to the reader to know how the sampling affects the discovery of germline sequences.

Response:

Random subsampling indeed influences somewhat which sequences are found by the cluster analysis, particularly in the beginning. However, the probability is large that all highly expressed sequences are represented in the random sample. Also, due to the database growing with subsequent iterations, the set of sequences assigned to a single database gene becomes smaller and more homogeneous. This makes it increasingly likely that also sequences expressed at lower levels result in a cluster since they now make up a larger fraction of each subsample.

We also observed that many of the clusters, which are captured in one subsample but not in the other are artifacts that were then filtered out anyway by the pre-germline or germline filter. To test this in practice, we conducted an experiment where we repeated a run of the program four times on the same human dataset, using identical parameters each time except that the subsampling was done in a different way. Although intermediate results differed, all four personalized databases that the program produced were exactly identical. Concordance is lower, though, when the input database is not as complete as the human one.

We have added the above text to the program's documentation.

9. Number reporting inconsistency

- The authors report 74 novel VH alleles identified for rhesus F132 (lines 369-370, Figure 8a). In Figure 8b, 71 novel VH alleles for F132 are reported instead.

- Figure 8d-e reports incongruent numbers between table and venn diagram (panel e).

F130 shows 47 total VK alleles in venn diagram while 49 are shown in Figure 8d. F124 shows 36 total VK alleles in venn diagram while 38 are shown in Figure 8d.

Response:

We thank the Reviewer for carefully going through the numbers in both figures and text. In the revised manuscript, the number of alleles reported has changed as a result of the new code added during the improvements of the IgDiscover program. This alters the output somewhat but does not change the overall conclusions. We now report that 62 new VH alleles were identified in animal F132 (Figure 8b and c), while 44 new VK alleles and 24 new VL alleles were found in F130 and 45 new VK alleles and 28 new VL alleles were found in F124. The numbers in the table (8b) and Venn diagrams (8c-e) as well as elsewhere in the paper are now congruent.

10. Figure 1b states that the Rhesus monkey has 106 alleles/genes. The authors claim in figure 5 that they used all of these alleles for IgDiscover (5A), but then only 80 alleles were returned. There does not appear to be an explicit statement of the ratio of VH genes to alleles,*

like what is done human and mouse. It would be important to know as to why only 80 alleles were returned, and if across all of the starting databases these same alleles were excluded. This would show some kind of systematic bias that would be important to mention.

Response:

There are three important points to consider here:

1. The number of 106 alleles/genes is derived from a combination of several previously published rhesus VH databases. Most of these sequences were produced by identifying possible candidate VH sequences present within the available rhesus genomic sequence and as such there is a distinct possibility that many of these sequences are non-functional (for example pseudogenic VH sequences).
2. Due to the issue of genomic assembly of short read sequences in highly duplicated genomic regions the rhesus genome, in common with many other species, does not have an uninterrupted reference sequence that completely encompasses the various Ig loci. As such there are many gaps in the current reference genome with the result that an unknown number of VH genes are not present in the current assembly.
3. The rhesus is a highly genetically diverse species. Comparison of the reference sequence of Indian and Chinese origin rhesus demonstrate a number of sequence differences in the VH genes between just these two classes of rhesus. As we have shown in the current study there is a significant level of diversity in terms of VH allelic sequences of multiple individual animals.

We have clarified the figure caption of Figure 4. 'Quality controlled' in this sentence refers only to the removal of sequences shorter than 300 bp. Using that term is a leftover from earlier versions of the pipeline in which we indeed had more elaborate filtering schemes, but since we lost too many usable sequences doing this, we now pass on nearly all sequences to IgBLAST, and perform almost most of the quality filtering after IgBLAST assignment. Details of this filtering, including thresholds, are described in Materials and Methods. We now refer the reader to Materials and Methods.

11. The authors should report how fast IgDiscover performs as a software platform. For example, the relationship of how fast de novo V-gene germlines are discovered starting from X number of NGS reads and X number of starting V-gene database should be included. This will help for future users to calibrate their datasets for implementation of the method.

Response:

IgDiscover running time depends, among other factors, on the number of reads that can be merged, the number of available CPU cores, and the number of requested iterations. We measured runtime on a human dataset with 1.4 million reads of good quality on an Intel Xeon E5-2660 with sixteen cores. Since the human database is already nearly complete, a single iteration is sufficient. In this best-case scenario, IgDiscover needs 95 minutes to discover a fully personalized V gene database. Pre-processing and the discovery process itself (including cluster analysis and consensus building) account for less than 10% of this time, whereas the remaining over 90% of the time is spent on running IgBLAST for obtaining gene

assignments.

In practice, IgDiscover runs three iterations by default to cover those cases where the input database is more distant to the analyzed sample. It also executes IgBLAST one additional time so that a gene usage profile based on the final personalized database can be computed. Runtime in this case, assuming again a 16-core CPU, is approximately five hours.

We have added these data to the paper as a supplementary section.

12. Lines 120-121, I suggest the authors refrain from describing '5 RACE as an "unbiased" amplification. Several papers have shown that substantial artifacts can occur during 5' RACE, (see: Cocquet et al., Genomics, 2006 and Best et al., Sci Reports, 2015). I understand that the authors probably mean that 5' RACE helps to avoid some of the biases and mispriming caused by multiplex-PCR, thus they should be more explicit to state this rather than simply calling 5' RACE unbiased.

Response:

Opinions vary on which amplification method produces the most unbiased set of libraries and whether 5'RACE is the best solution to this issue. Because we value the IgDiscover process and want to make it available to as many people as possible we have worked to develop a multiplex PCR library approach that enables libraries to be prepared from species that are better characterized – in this case human and mouse. Additionally we have altered the text as the reviewer suggested to remove the remark about 5'RACE being unbiased.

Importantly, the IgDiscover program can process NGS antibody sequences independently of the library production method or the sequencing platform. With the free availability of multiple archived antibody sequence datasets, IgDiscover will enable researchers that may lack the resources to produce their own libraries to perform germline discovery and analysis from archived material produced for previous studies.

13. Throughout the manuscript the authors should be more specific about which mouse strain they are using or referring to. For example, only the C57BL6 mouse strain has a fully sequenced and annotated genome and immunoglobulin locus. This not however the case for Balb/c.

Response:

The original data used a C57BL6 mouse. We have subsequently created three new libraries for the paper – two from balb/c mice and one from C57BL6. We are now careful to mention this in the manuscript. We should point out, however, that even with inbred laboratory strains there will be some degree of genetic diversity and that, coupled with the fact that the Ig locus is the site of frequent segmental duplications, means that the current C56BL6 reference sequence will not be definitive for all members of this strain. It is likely that additional novel sequences may be present in other C56BL6 mice.

In addition we demonstrate that IgDiscover identify germline sequences from archived NGS sequence data – in this case a mouse IgM library prepared on the Ion Torrent sequencing platform.

Reviewer #2

This paper describes a new computational method to infer germline immune gene segments, based on next gen immune repertoire sequencing. The paper also suggests a message for personalized databases of immune repertoire genes. The topic is timely, since this field of immune repertoire sequencing is exploding, both in humans and experimental animals, but the pipelines to analyze these sequence sets, and the databases of gene segments with which to align sequences is incomplete or missing. The method described here offers a way to fill that gap until genomic gene segments are fully defined, and also offers a personalized way to analyze the repertoires of individual people or animals with polymorphisms.

The paper is very dense. The iterative nature of the computational work and the lab validation is important, but complex to follow. Given it is essentially a methods paper, I don't have any specific suggestions to decomplexify the paper, since aficionados who will try to use this method will want to see the detail.

The data is well described, and the methods for both computation and lab seem state of the art. All details needed to reproduce the lab work are provided, and the code has been made accessible.

No doubt there is some fuzziness in the assignment of gene segment numbers or sequences in this paper, but that is the nature of the work at this point, and is ok.

Response:

We thank the Reviewer for their positive remarks about our paper.

To make the overall IgDiscover process clearer we have now added an additional supplementary figure that gives an overview of the process with an emphasis on the novel aspects of the technique, specifically comparative analysis, database replacement, iterative database building, whitelisting and germline filter steps. We agree with Reviewer 2 regarding the gene assignment numbers. The assignment of an official nomenclature to the discovered alleles is a separate issue from the process of gene discovery and we assume the designated names for alleles discovered in this study can be changed by an official nomenclature body.

Responses to reviewers

Reviewer #1

“I thank the authors for the thorough and thoughtful responses to my questions. They have done an excellent job in revising the manuscript and improving the few areas of concern.”

We thank reviewer #1 for their comment and also would like to thank them for their previous inciteful comments about IgDiscover, the addressing of which have led to a greatly improved final manuscript and program.

Reviewer #2

The response to the previous reviewer queries appears thorough and improves the manuscript, especially the reconstruction of nearly complete gene sets.

We thank reviewer #2 for their positive comment and agree that the reconstruction of these gene sets greatly improves the manuscript.